# Sample Average Approximation for Black-Box Variational Inference

## Abstract

We present a novel approach for black-box VI that bypasses the difficulties of stochastic gradient ascent, including the task of selecting step-sizes. Our approach involves using a sequence of sample average approximation (SAA) problems. SAA approximates the solution of stochastic optimization problems by transforming them into deterministic ones. We use quasi-Newton methods and line search to solve each deterministic optimization problem and present a heuristic policy to automate hyperparameter selection. Our experiments show that our method simplifies the VI problem and achieves faster performance than existing methods.

## 1 Introduction

The goal of probabilistic inference is to approximate the distribution of latent variables given observed data and a user-specified model. This is of immense practical interest in fields like astrophysics, epidemiology, political science, psychology, ecology, and others. Typically this is used in applications where the data are limited so that user-provided assumptions are critical to obtain insight from data. However, inference is extremely challenging in general and formally computationally intractable except for restricted cases. It is therefore a long-standing research direction to develop robust approximate inference methods that perform well on a wide range of real models.

Variational inference (VI) is a powerful technique that allows us to approximate the posterior distribution by formulating inference as an optimization problem, where the objective is to find a distribution from a family of distributions that is as close as possible to the true distribution. To achieve this, VI maximizes the evidence lower bound (ELBO), which is a lower bound on the log-likelihood of the observed data [Wainwright et al., 2008; Jaakkola and Jordan, 1997; Beal, 2003].

Much recent work focuses on "black box" variational inference (BBVI) [Ranganath et al., 2014; Titsias and Lázaro-Gredilla, 2014; Kucukelbir et al., 2017; Yin and Zhou, 2018; Hoffman and Ma, 2020; Buchholz et al., 2018], where the goal is to adapt a tractable density to approximate the posterior using only evaluations of the model's log-joint density or its gradient. This allows VI to be applied to a wide range of models, for example, those written in a probabilistic programming language such as Stan [Carpenter et al., 2017]. BBVI uses stochastic gradient descent (SGD) to optimize the ELBO [Wingate and Weber, 2013; Blei et al., 2017; Kucukelbir et al., 2017; Ranganath et al., 2014; Rezende et al., 2014; Kingma and Welling, 2013]. In this method, an unbiased estimator of the gradient of the ELBO is used for stochastic optimization. However, selecting an appropriate step-size for SGD can be challenging and have a significant impact on the optimization process's outcome. Recent work by Agrawal et al. [2020] recommends performing a comprehensive search over step-sizes to avoid the suboptimality of using a previously-selected step-size. In practice, users often turn to adaptive methods like Adam [Kingma and Ba, 2015] or AdaGrad [Duchi et al., 2011] to adjust the step-size on-the-fly, but these methods also require tuning of hyperparameters, which can be time-consuming and error-prone.

In principle, BBVI simplifies an inference problem by converting it to stochastic optimization. However, in practice the difficulties of stochastic gradient methods have limited its robustness and broad applicability [Agrawal et al., 2020; Welandawe et al., 2022]. This motivates the consideration of alternate stochastic

optimization methods that perform more reliably for classes of BBVI problems. We propose an alternative stochastic optimization approach based on the sample average approximation (SAA) that can be easily made robust for problems involving hundreds of latent variables, specifically focusing on statistical models that do not rely on data-subsampling. Our main contributions are:

(i) We propose using SAA to provide solutions for the variational inference problem. SAA approximates the expectation of the objective function by the sample average using a fixed sample from the distribution [Healy and Schruben, 1991; Robinson, 1996; Shapiro and Wardi, 1996; Kleywegt et al., 2002; Kim et al., 2015]. This deterministic optimization problem allows us to employ nonlinear optimization tools usually inaccessible for this task.

(ii) Our novel approach applies quasi-Newton methods with line-search to solve the deterministic optimization problem resulting from SAA. This method emphasizes efficient optimization without manual step-size tuning and takes advantage of the deterministic nature of the SAA problem. By employing SAA, we reduce the number of iterations while using larger sample sizes in each iteration to compute the probabilistic model and its approximation. This approach enhances optimization efficiency by leveraging the vectorization capabilities of GPUs.

(iii) To address the Monte Carlo error introduced by SAA, we propose using retrospective approximation [Chen and Schmeiser, 2001], a technique that improves SAA's accuracy by employing a sequence of SAAs with increasing sample sizes.[1]

(iv) We present the SAA for VI algorithm, which includes default scheduling for sample sizes and a stopping criterion. Our empirical results demonstrate that our approach is competitive with state-of-the-art methods, including the batched quasi-Newton method of Liu and Owen [2021], in terms of accuracy and computational cost, and has the potential to simplify the variational inference process.

Concurrently with our work, Giordano et al. [2023] proposed a sample average approximation algorithm for variational inference, motivated by the same challenges of stochastic gradient methods that limit the robustness and broad applicability of BBVI. We discuss the relationship between our method and theirs in Section 4.

## 2 Background

We are interested in approximating the posterior distribution of a latent variable given some observed data, i.e., $p(Z \mid x)$, where $Z$ is the latent variable and $x$ is the observed data. To achieve this, we will approximate the posterior with a distribution from an indexed family of approximations $\mathcal{Q} = \{q_\theta \mid \theta \in \mathbb{R}^d\}$, where $\theta$ is a vector of parameters that parameterize the approximation $q_\theta(Z)$, and $d$ is the dimension of $\theta$.

VI proposes to approximate the posterior distribution by finding a member from $\mathcal{Q}$ that is closest in Kullback-Leibler divergence to the true distribution. This is achieved by maximizing the evidence lower bound (ELBO), which is a function of the parameters $\theta$:

$$\mathcal{L}(\theta) = \mathbb{E}[\ln p(Z, x) - \ln q_\theta(Z)], \qquad Z \sim q_\theta. \qquad (1)$$

The optimization problem can be formulated as:

$$\max_{\theta \in \Theta} \mathcal{L}(\theta) = \max_{\theta \in \Theta} \mathbb{E}[\ln p(Z, x) - \ln q_\theta(Z)], \qquad Z \sim q_\theta. \qquad (2)$$

Under smoothness assumptions, black-box VI presents this problem as a smooth stochastic optimization problem (SOP) and suggests solving it using methods based on stochastic gradient descent (SGD). Specifically, it uses stochastic gradient ascent to maximize the ELBO by updating the parameters as follows:

---

[1]See Emelogu et al. [2016] for a literature review on the topic.

At every iteration, samples $z_1, \ldots, z_n$ from $q_{\theta_t}$ are drawn and the sample mean of the function $g_{\theta_t}(Z)$ is being computed, where $g_{\theta_t}(Z)$ is a $\mathbb{R}^d$-valued random vector whose expectation equals the gradient. Then, this estimate is used to update the parameters according to:

$$\theta_{t+1} = \theta_t + \gamma_t \frac{1}{n} \sum_{i=1}^{n} g_{\theta_t}(z_i) \qquad \text{for } t \in \mathbb{N}, \text{ and } \gamma_t \in \mathbb{R}+. \tag{3}$$

This function can be obtained using various methods, including the score function estimator [Wingate and Weber, 2013; Ranganath et al., 2014] or, if the distribution is reparameterizable, the 'reparameterization trick' [Kingma and Welling, 2013; Fu, 2006; Kingma et al., 2019; Rezende et al., 2014], among others. A random variable $Z$ comes from a reparameterizable distribution $q_\theta$ if there exist a $C^1$ function $z_\theta$ and a density $q_{\text{base}}$ such that $Z = z_\theta(\epsilon)$ for $\epsilon \sim q_{\text{base}}$. We refer to these $\epsilon$ values as noise. In such case, the stochastic optimization problem becomes

$$\max_{\theta \in \Theta} \mathcal{L}(\theta) = \max_{\theta \in \Theta} \mathbb{E}[\ln p(z_\theta(\epsilon), x) - \ln q_\theta(z_\theta(\epsilon))], \qquad \epsilon \sim q_{\text{base}}. \tag{4}$$

It then follows that, at every step $t$ of the optimization, the update rule of SGD from Eq. (3) is

$$\theta_{t+1} = \theta_t + \gamma_t \frac{1}{n} \sum_{i=1}^{n} g_{\theta_t}(z_{\theta_t}(\epsilon_{ti})), \qquad \epsilon_{ti} \sim q_{\text{base}}.$$

Despite its simplicity, the explanation above fails to convey the complexities of choosing hyperparameters, particularly the step size $\gamma_t$, also known as the learning rate. The user can opt to use a step size schedule $\boldsymbol{\gamma} = (\gamma_t)_{t \in \mathbb{N}} \subset \mathbb{R}_+$ that meets the Robbins-Monro conditions ($\|\boldsymbol{\gamma}\|_1 = \infty$ and $\|\boldsymbol{\gamma}\|_2 < \infty$), which can lead to SGD converging at a critical point due to the use of unbiased estimators of the gradients [Robbins and Monro, 1951; Ranganath et al., 2014; Jankowiak and Obermeyer, 2018]. However, the specific sequence of the schedule is not specified and different schedules may affect the speed of convergence differently [cf. Agrawal et al. [2020]]. Critically, the random nature of estimating the loss function and its gradient makes it impractical to use traditional line-search methods. Additionally, the choice of the number of samples $n$ drawn at each iteration can affect the optimization process, as a larger $n$ provides a more accurate gradient estimate but may increase the computational cost. Balancing this trade-off is an important aspect of algorithm design.

Moreover, controlling the variance of gradient estimates significantly influences the performance of the optimization algorithm, affecting stability and convergence properties, and further adding to the complexity of the problem. In this context, the choice of the gradient estimator $g_{\theta_t}$ is crucial. Instead of employing the naïve estimator by taking the average of the gradient of $\ln p(z_{\theta_t}(\epsilon)) - \ln q_{\theta_t}(z_{\theta_t}(\epsilon))$, one can consider alternative methods such as the sticking-the-landing estimator [Roeder et al., 2017] or, when the entropy term $\mathbb{H}_\theta = -\mathbb{E}[\ln q_{\theta_t}(z_{\theta_t}(\epsilon))]$ is available in closed form, estimating the gradients of $\mathbb{E}[\ln p(z_{\theta_t}(\epsilon))] + \mathbb{H}_\theta$. Although all these estimators are unbiased, they exhibit different variance behaviors, which can impact the optimization process. To reduce the variance of the gradient estimator, control variates can also be applied [Ranganath et al., 2014; Geffner and Domke, 2018]. These choices contribute to the overall complexity of choosing hyperparameters, step size schedules, and the number of samples.

## 3 Methods

### 3.1 Sample Average Approximation

The problem of ELBO maximization in the reparameterization setting of Eq. (4) is formulated as an SOP where the stochasticity comes from a fixed probability distribution, i.e., a probability distribution which does not depend on $\theta$. Furthermore, the function inside the expectation is a smooth function of the parameters $\theta$. Solutions to these problems can be approximated using the *sample average approximation* (SAA): a sample average over a *fixed sample* replaces the expectation, effectively transforming the SOP into a deterministic optimization problem.

We propose to use SAA for black-box VI. To use SAA, we take $n$ i.i.d. samples $\boldsymbol{\epsilon} = \epsilon_1, \ldots, \epsilon_n$ from the distribution $q_{\text{base}}$ and define the deterministic *training objective* function

$$\hat{\mathcal{L}}_{\boldsymbol{\epsilon}} \colon \theta \mapsto \frac{1}{n} \sum_{i=1}^{n} [\ln p(z_\theta(\epsilon_i), x) - \ln q_\theta(z_\theta(\epsilon_i))],$$

which is a function of $\theta$ alone.

Then, the optimization problem in Eq. (4) can be transformed into a deterministic optimization problem

$$\max_{\theta \in \Theta} \hat{\mathcal{L}}_{\boldsymbol{\epsilon}}(\theta) = \max_{\theta \in \Theta} \frac{1}{n} \sum_{i=1}^{n} [\ln p(z_\theta(\epsilon_i), x) - \ln q_\theta(z_\theta(\epsilon_i))] = \max_{\theta \in \Theta} \frac{1}{n} \sum_{i=1}^{n} v_\theta(\epsilon_i), \qquad (5)$$

where we introduced the *log-weights* $v_\theta(\epsilon_i) = \ln p(z_\theta(\epsilon_i), x) - \ln q_\theta(z_\theta(\epsilon_i))$, also known as *log-importance ratios*. As the optimization is performed with the fixed set $\boldsymbol{\epsilon}$, we refer to it as the training noise.

We want to recover the optimal parameters $\theta^*$ of $\hat{\mathcal{L}}_{\boldsymbol{\epsilon}}$. In an unconstrained smooth optimization setting, we need to specify how to compute a search direction and a step size. For the search direction, we will use L-BFGS [Broyden, 1970; Fletcher, 2013; Goldfarb, 1970; Shanno, 1970; Nocedal, 1980]. For a detailed description of the L-BFGS algorithm, refer to Nocedal and Wright [1999].

In contrast to the SGD setting, deterministic optimization allows us to specify the step size using line search and ask for it to satisfy the *strong Wolfe conditions* [Nocedal and Wright, 1999]. Specifically, for $0 < c_1 < c_2 < 1$, the step size $\gamma$ must simultaneously satisfy the following two conditions:

    i) sufficient increase: $\hat{\mathcal{L}}_{\boldsymbol{\epsilon}}(\theta + \gamma \mathbf{r}) \geq \hat{\mathcal{L}}_{\boldsymbol{\epsilon}}(\theta) + c_1 \gamma \nabla \hat{\mathcal{L}}_{\boldsymbol{\epsilon}}(\theta)^{\mathrm{T}} \mathbf{r}$, and

    ii) modified curvature: $\left| \nabla \hat{\mathcal{L}}_{\boldsymbol{\epsilon}}(\theta + \gamma \mathbf{r})^{\mathrm{T}} \mathbf{r} \right| \leq c_2 \left| \nabla \hat{\mathcal{L}}_{\boldsymbol{\epsilon}}(\theta)^{\mathrm{T}} \mathbf{r} \right|$.

We will use L-BFGS with line search to find a local optimum of Eq. (5), and denote the process that does so by $\mathrm{Opt}(\theta, n, \boldsymbol{\epsilon}, \tau)$. Here, $\tau$ is the maximum number of iterations for which L-BFGS will run, and $\theta$ is an initial value of the parameters. Besides the arguments of $\hat{\mathcal{L}}_{\boldsymbol{\epsilon}}(\theta)$, we also need to specify the value of $\tau$.

**Detection and mitigation of overfitting.** It is important to understand that the training objective $\hat{\mathcal{L}}_{\boldsymbol{\epsilon}}(\theta)$ and the ELBO $\mathcal{L}(\theta)$, may differ for a fixed $\theta$. The ELBO, as defined in Eq. (1), is an expectation over the distribution $q_\theta$, while the training objective is computed based on an average over a fixed sample $\boldsymbol{\epsilon}$. In contrast, the optimal ELBO refers to the value of the ELBO achieved by the maximizer of Eq. (2), denoted as $\theta^*$.

During optimization with a fixed sample of training noise $\boldsymbol{\epsilon}_n = \epsilon_1, \ldots, \epsilon_n$, one might wonder how much the learned parameters $\theta^*_{\boldsymbol{\epsilon}_n}$ and the distribution $q_{\theta^*_{\boldsymbol{\epsilon}_n}}$ depend on these noise samples. In particular, how this dependency translates into a gap between the ELBO $\mathcal{L}(\theta^*_{\boldsymbol{\epsilon}_n})$ and the optimal ELBO $\mathcal{L}(\theta^*)$. Fortunately, there are two results by Mak et al. [1999] that are relevant to our discussion. Note that until the noise variables $\epsilon_1, \ldots, \epsilon_n$ are realized, the quantities $\theta^*_{\boldsymbol{\epsilon}_n}$ and $\hat{\mathcal{L}}_{\boldsymbol{\epsilon}_n}(\theta^*_{\boldsymbol{\epsilon}_n})$ are random. Let $\hat{\boldsymbol{\epsilon}}_{n+1} = \hat{\epsilon}_1, \ldots, \hat{\epsilon}_{n+1}$, be a sample of size $n + 1$ taken i.i.d. from $q_{\text{base}}$. Assuming that the optimization process converges to a global optimum, it holds that: (i) the ELBO and training objective sandwich the optimal ELBO (in expectation), that is, $\mathbb{E}\,\mathcal{L}(\theta^*_{\boldsymbol{\epsilon}_n}) \leq \mathcal{L}(\theta^*) \leq \mathbb{E}\,\hat{\mathcal{L}}_{\boldsymbol{\epsilon}_n}(\theta^*_{\boldsymbol{\epsilon}_n})$; and (ii) the training objective converges monotonically to the optimal ELBO from above (in expectation), that is, $\mathbb{E}\,\hat{\mathcal{L}}_{\hat{\boldsymbol{\epsilon}}_{n+1}}(\theta^*_{\hat{\boldsymbol{\epsilon}}_{n+1}}) \leq \mathbb{E}\,\hat{\mathcal{L}}_{\boldsymbol{\epsilon}_n}(\theta^*_{\boldsymbol{\epsilon}_n})$.

In particular, these results mean that we can use standard statistical techniques to quantify the discrepancy between the ELBO $\mathcal{L}(\theta^*)$ and the training objective $\hat{\mathcal{L}}_{\boldsymbol{\epsilon}}(\theta)$ by comparing the distribution of the log-weights $v_1, \ldots, v_n$ for a fresh sample of noise, referred to as testing noise, and the training noise, a technique first used by Mak et al. [1999]. Figure 1 displays the distribution of log-weights for a growing sample size. As the number of samples increases, the training objective value decreases and approaches that of the ELBO estimation, which in turn increases, indicating progress toward the ultimate goal of ELBO maximization.

We adopt the classical approach of mitigating this overfitting by solving a sequence of SAA approximations for an increasing sequence of sample sizes $(n_t)_{t \in \mathbb{N}} \subseteq \mathbb{N}$, which creates a sequence of solutions $(\theta^*_{n_t})_{t \in \mathbb{N}}$.

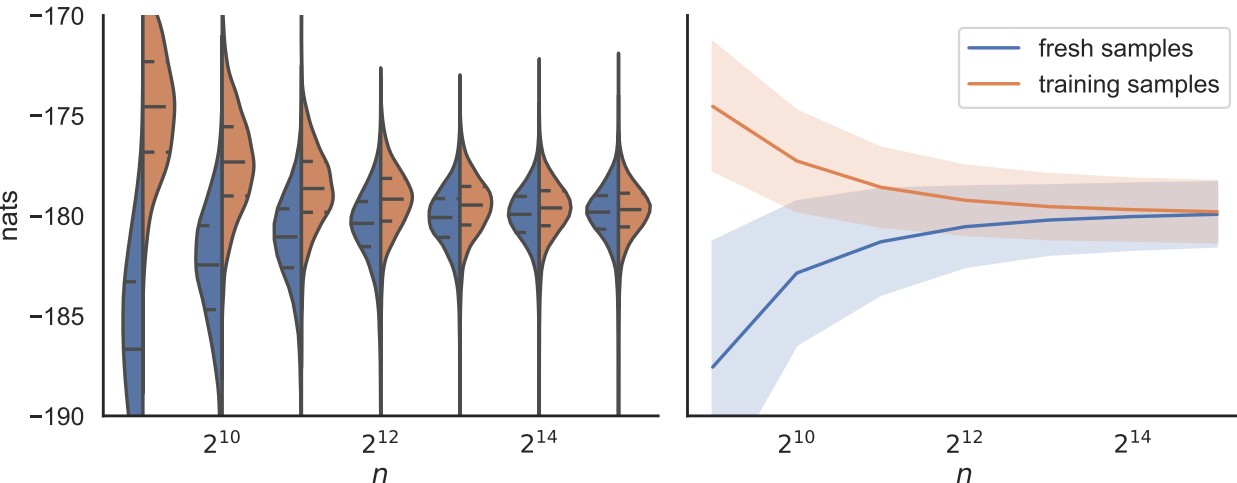

Figure 1: Distribution of log-weights for a fresh sample of noise and the training noise, as a function of the number $n$ of samples used for training on the `mushrooms` dataset. (Left) Violin plot showing the distribution of log-weights. (Right) Line plot depicting the mean ($\bar{v} \pm \sigma$) of the log-weights. The means of the log-weights correspond to an estimation of the ELBO and the training objective. The overfitting to the training noise is reduced by training using a larger sample size.

Shapiro [2003] give general conditions for the set of optimal solutions (or critical points) of SAA problems to converge to the corresponding set for the original stochastic optimization problem. The conditions include uniform convergence of the SAA objective functions and compactness of the solution set (see also Kim et al. [2015]). While these could likely be applied to VI problems, the conditions, especially compactness of the solution set, would be problem specific and depend, for example, on the particular parameterization of a variational distribution, and we don't explore it further.

## 3.2  Algorithm

In this section, we present an algorithm that uses SAA to approximate the solution to the optimization problem of maximizing the ELBO. Our objective is to find a good approximation to the solution with a reasonable computational cost and avoid the overfitting phenomenon described earlier. To this end, we build our stopping criteria based on overfitting. The algorithm, described in Algorithm 1, consists of two procedures: the optimizer Opt and the convergence checker. We previously described the optimizer, in which we used a quasi-Newton method. The convergence checker is a function that determines whether we need to continue the optimization process, and we will describe it later in this section.

The algorithm starts with an initial guess $\theta_0$, an initial sample size $n_0$, and an initial maximum number of iterations for the optimizer $\tau_0$. Default values for these parameters can be found in Table 1. At each iteration $t$ of the algorithm, we double the sample size to reduce the overfitting. First, we draw the training noise $\boldsymbol{\epsilon}_{n_t} = \epsilon_1, \ldots, \epsilon_{n_t}$ from the base distribution $q_{\text{base}}$. Then, we use the optimizer to find the parameters $\theta_t^*$ that maximize the deterministic objective computed with the fixed training noise $\boldsymbol{\epsilon}_{n_t}$. If we find that the optimizer has reached the maximum number of iterations $\tau_t$, we double the maximum number of iterations for the optimizer; otherwise, we keep the same maximum number of iterations.

We track a counter variable `count`, which increments each time the optimizer Opt uses less than a very small number of iterations, `VERY_SMALL_ITER`. If the `count` variable reaches 3, it means that the optimizer has finished with a very small number of iterations for three consecutive step sizes, and the optimization process terminates. Otherwise, we reset `count` to 0 if the optimizer uses more than `VERY_SMALL_ITER` iterations. The optimization process continues until either the convergence checker determines that we have reached a good solution to the stochastic optimization problem or the `count` variable reaches 3.

**Algorithm 1** SAA for VI

1: **Input:** $\theta$, $n$, $\tau$
2: **Output:** $\theta^*$
3: $t \leftarrow 0$, count $\leftarrow 0$
4: **while** count $< 3$ **do**
5:     $t \leftarrow t + 1$, $n \leftarrow 2n$
6:     $\boldsymbol{\epsilon}_n \leftarrow \epsilon_1, \ldots, \epsilon_n$, where $\epsilon_i \sim q_{\text{base}}$
7:     $\theta \leftarrow \text{Opt}(\theta, n, \boldsymbol{\epsilon}_n, \tau)$
8:     $\eta \leftarrow$ number of iter used by the optimizer
9:     **if** $\eta = \tau$ **then**
10:         $\tau \leftarrow 2\tau$
11:     **if** $\eta < \text{VERY\_SMALL\_ITER}$ **then**
12:         count $\leftarrow$ count $+ 1$
13:     **else**
14:         count $\leftarrow 0$
15:     **if** count $= 0$ **and** converged$(\theta, \boldsymbol{\epsilon}_n, t)$ **then**
16:         **break**
17: **return** $\theta^* \leftarrow \theta$

**Algorithm 2** converged?

1: **Input:** $\theta$, $\boldsymbol{\epsilon}_n$, $t$
2: **Params:** max\_t, $\delta$
3: **Output:** converged, a boolean
4: converged $\leftarrow$ False
5: $\hat{\boldsymbol{\epsilon}}_{10\text{k}} \leftarrow \hat{\epsilon}_1, \ldots, \hat{\epsilon}_{10\text{k}}$, where $\hat{\epsilon}_i \sim q_{\text{base}}$
6: obj $\leftarrow \text{mean}(v_\theta(\boldsymbol{\epsilon}_n))$
7: elbo $\leftarrow \text{mean}(v_\theta(\hat{\boldsymbol{\epsilon}}_{10\text{k}}))$
    ▷ Perform statistical test using t-tests:
8: $\text{p}_{\text{value}} \leftarrow \text{t\_test}(v_\theta(\boldsymbol{\epsilon}_n), v_\theta(\hat{\boldsymbol{\epsilon}}_{10\text{k}}))$
9: **if** $\text{p}_{\text{value}} > 0.01$ **then**
10:     converged $\leftarrow$ True
11: **if** $|\text{obj} - \text{elbo}| < \delta$ **or** $t \geq$ max\_t **then**
12:     converged $\leftarrow$ True
13: **return** converged

| Input | Default value |
|---|---|
| $\theta$ | random from $\mathcal{N}(0,1)$ |
| $n$ | 32 |
| $\tau$ | 300 |

Table 1: Default values for the input parameters of SAA for VI.

**Stopping** Algorithm 2 defines the stopping criteria for our optimization process, which involves computing log-weights. Specifically, given the training noise $\boldsymbol{\epsilon}_{n_t}$ and the parameters $\theta_t$, we compute the log-weights $v_{\theta_t}(\epsilon_1), \ldots, v_{\theta_t}(\epsilon_{n_t})$, which we denote as $v_{\theta_t}(\boldsymbol{\epsilon}_{n_t})$. We also compute a new set of log-weights using a fresh sample of testing noise with size 10k, denoted by $v_{\theta_t}(\hat{\boldsymbol{\epsilon}}_{10\text{k}})$.

To decide when to stop optimizing, we use a two-sided t-test to compare the distribution of log-weights computed using the training noise $v_{\theta_t}(\boldsymbol{\epsilon}_{n_t})$ with the distribution of log-weights computed using the testing noise $v_{\theta_t}(\hat{\boldsymbol{\epsilon}}_{10\text{k}})$. The null hypothesis is that the means of the two distributions are the same. Although the assumptions for the test (e.g., that the training log-weights are i.i.d.) may not necessarily hold, we utilize the test as a heuristic to determine when to stop. This approach provides a threshold for the difference between the training objective and the ELBO estimation that remains robust to the noise present in the log-weights. The test is inspired by the one used in Mak et al. [1999].

Our optimization process terminates when the null hypothesis cannot be rejected with a significance level of 0.01%. Note that the convergence check is only performed if `count = 0`. This ensures that we test for convergence only when the optimizer has made some updates. Otherwise, the convergence check would be meaningless, since the distribution of the training log-weights $v_{\theta_t}(\boldsymbol{\epsilon}_{n_t})$ and the testing log-weights $v_{\theta_t}(\hat{\boldsymbol{\epsilon}}_{10\text{k}})$ would be very similar. We also introduce two additional stopping conditions: the maximum number of iterations max\_t and the threshold $\delta$ for the difference between the training objective $\hat{\mathcal{L}}_{\boldsymbol{\epsilon}}(\theta_t)$ and the ELBO $\mathcal{L}(\theta_t)$. In our experiments, we set max\_t to ensure that the maximum sample size was $n_{\max} = 2^{18}$, and $\delta$ to 0.01. In Appendix F, we provide a more detailed discussion of the hyperparameters used in our experiments.

## 4 Related work

In the existing literature, there are efforts to incorporate second-order information into stochastic optimization, which have been applied to VI. Byrd et al. [2016] introduced a method that employs the L-BFGS update formula through subsampled Hessian-vector products, referred to as batched-L-BFGS or batched quasi-Newton. Liu and Owen [2021] applied the method from Byrd et al. [2016] to address the variational inference problem, with the optional inclusion of quasi-Monte Carlo (QMC) sampling to further decrease the variance of the gradient estimator. Both approaches involve a two-step algorithm: (1) updating the parameters at each iteration using L-BFGS's two-loop recursion, and (2) updating the displacement vector **s** and gradient difference vector **y** of L-BFGS every $B$ steps by employing the average of the parameters from the preceding $B$ iterations. In the work of Liu and Owen [2021], each iteration involves drawing a fixed-size sample of noise $\epsilon$ from $q_{\text{base}}$ to estimate the ELBO gradient and conduct the line search. The sample size is not extensively discussed in their work; however, the experiments were conducted with sample sizes of 128 or 256. These values are larger than those typically used in the literature, suggesting that the sample size could indeed be a relevant factor to consider. Our method deviates from the approach proposed by Liu and Owen [2021] in two key ways. Firstly, we execute a complete deterministic optimization using a fixed set of noise, effectively reducing uncertainty. Secondly, we seamlessly integrate the sample size consideration into the algorithm itself, consequently minimizing the need for user input. As we demonstrate in Section 5.1.2, these differences lead to significant improvements when handling complex target and approximating distributions.

An alternative approach to incorporating second-order information into the variational inference problem can be found in the work of Zhang et al. [2022]. Their method employs L-BFGS to identify modes or poles of the posterior distribution. Subsequently, the data generated by L-BFGS is utilized to estimate the posterior covariance around the mode, which is then used to parameterize an approximating distribution. This approach more closely resembles the Laplace approximation than methods that seek approximations to a global optimizer of the ELBO from a fixed parametric family.

We share a common goal with Welandawe et al. [2022], who also drew inspiration from Agrawal et al. [2020] to develop a system for variational inference that requires minimal user input. However, their method employs SGD for optimizing the ELBO and uses a heuristic schedule to update the step size $\gamma_t$ during the optimization process. They initially use a fixed step size and incorporate tools to detect when the SGD process reaches stationarity, at which point they decrease the step size by a factor $\rho$. During the stationary regime, they calculate the average of the parameters and take it as the optimal parameters for a given step size $\theta^*_{\gamma_t}$. They repeat the process of decreasing the step size until the symmetrized KL divergence between the current distribution and the optimal distribution $q_*$ (for the approximating family) falls below a threshold $\xi$. Notably, since the optimal distribution $q_*$ is not known, the authors estimated the KL divergence between $q_*$ and the current distribution $q_{\theta^*_{\gamma_t}}$. The authors observed that taking the average of the parameters in the stationary regime significantly improves the approximation quality compared to considering each parameter at every iteration. Although not directly related, our work shares with AutoML [He et al., 2021; Zöller and Huber, 2021] the desire of reducing user intervention. However, instead of a very broad scope, our work focus on the variational inference task, where the model and approximating families are given.

In the machine learning literature, the application of sample average approximation has been relatively rare. Some early works include PEGASUS by Ng and Jordan [2000], in which the authors addressed partially observable Markov decision processes by replacing the *value of a policy* (an expectation) with the sample average of the value function applied to a finite number of states for optimization purposes. In a different context, Sheldon et al. [2010] explicitly utilized the sample average approximation technique in a network design setting, where a naïve greedy approach was not applicable. More recently, Balandat et al. [2020] adopted sample average approximation to optimize the acquisition function in Bayesian optimization. SAA was previously used for VI in a specialized capacity in several papers [Giordano et al., 2018; Domke and Sheldon, 2018; Giordano et al., 2019; Domke and Sheldon, 2019; Giordano et al., 2022]; our work and the concurrent work of Giordano et al. [2023] are the first to explore its general applicability.

As mentioned in the introduction, Giordano et al. [2023] concurrently and independently developed a method based on the sample average approximation for black-box variational inference. The two papers employ the same basic algorithmic idea but have several differences in scope. Unlike Giordano et al. [2023], we focus

substantially on the case where SAA with a fixed sample size has significant error and therefore one needs to solve a sequence of problems with increasing sample sizes. We introduce heuristics that guide the selection of sample sizes and the decision of when to halt the process. On the other hand, Giordano et al. [2023] exploit the determinism of the SAA problem to develop techniques based on sensitivity analysis and the theory of "linear response covariances" [Giordano et al., 2015; 2018] to improve posterior covariance estimates of black-box VI and to estimate the Monte Carlo error of the SAA procedure, which are outside the scope of our work. They also present a theoretical result about a failure mode for SAA with too few samples relative to the dimension of the latent variables: specifically, for a Gaussian approximation with a dense covariance matrix, the sample size $n$ needs to be at least equal to the dimension $d$ of the latent space for the SAA problem to be bounded. Interestingly, while they conclude that this precludes using SAA for VI with a full Gaussian approximation, we show in the experiments section that, for interesting models, it is indeed feasible. Two explanations are that: (1) we consider sequences of SAA problems with sample sizes that can grow substantially larger (up to $2^{18}$) than they consider (usually 30), (2) our largest model has 501 latent variables, while they examine three models with larger sizes (up to 15K). Thus, their theoretical result provides useful guidance on the limitations of SAA for VI, while our empirical work shows that SAA for VI can be practical up to quite large sample sizes. Finally, we have provided an addendum in Appendix E that uses their theoretical result to improve our method: when using a dense approximation, the sequence of SAA problems should begin with a sample size larger then $d$; this makes SAA for VI even faster by avoiding wasted effort for small sample sizes.

## 5    Experiments

In this section, we present experimental evidence for our proposed method. We adopt the experimental setup of Burroni et al. [2023] and consider two types of models: 11 models from the Stan examples repository [Stan Development Team, 2021; Carpenter et al., 2017] and Bayesian logistic regression with 6 UCI datasets [Dua and Graff, 2017]. For each model $p(\mathbf{Z}, x)$, where $\mathbf{Z}$ is a $d$-dimensional random vector, the approximating distribution $q_\theta$ can either be a diagonal Gaussian or a $d$-dimensional multivariate Gaussian distribution. The former is a product of $d$ independent Gaussians, where the parameters $\mu_i$ and $\sigma_i^2 > 0$ are specific to each $Z_i$. The latter has parameters $\mu_i$ and $LL^{\mathrm{T}}$, where $L \in \mathbb{R}^{d \times d}$ is a lower-triangular matrix with diagonal elements that are positive, enforced by applying the `softplus` transformation. We use the constraints framework from PyTorch [Paszke et al., 2019] to transform the model $p$ into one with unconstrained real-valued latent variables, as done by Kucukelbir et al. [2017]. We run all our experiments on GPUs.

We run two sets of experiments. First, we conduct performance comparisons where we assess our proposed method against two other methods: Adam with a fixed step-size, which is commonly used for black-box VI optimization, and batched quasi-Newton, a newer method that introduces second-order information in the optimization process. For all methods compared, we employed the naïve gradient estimator described in Section 2. When using Gaussian approximating distributions, this estimator corresponds to the one obtained when the entropy term is computed in closed-form. Second, we conduct an ablation study to explore how our decisions affect the algorithm's performance. We present the results of these experiments in the following subsections.

### 5.1    Performance comparison

#### 5.1.1    Adam

In order to solve the black-box VI problem, it is standard practice to use Adam [Kingma and Ba, 2015] as the default optimizer. This is evident from examples in Pyro[2] and the TensorFlow-Probability VI tutorial.[3] Despite the fact that the influence of the step-size in the optimization process is less relevant with Adam than with SGD, it is still a factor to consider. In our study, we compared Adam to our proposed method, SAA for VI. For Adam, we optimized each model and approximating distribution combination with three different step-sizes: 0.1, 0.01, and 0.001, and 20 repetitions of each combination. At each iteration with Adam, we

---

[2]See, for instance, the examples in Pyro-SVI.
[3]Adam is also used in the TensorFlow-Probability VI Tutorial.

estimated the gradient of the ELBO by taking 16 samples from $q_\theta$. For each model and approximating distribution, we selected the step-size that provided the highest median ELBO across the 20 repetitions. Please see Appendix A for more details on the Adam experiments. For SAA for VI, we used the algorithm described in Section 3.2, using the default parameter values of Table 1.

| | Diagonal Covariance | | | Dense Covariance | | |
|---|---|---|---|---|---|---|
| | Adam | SAA for VI | Difference | Adam | SAA for VI | Difference |
| | (i) | (ii) | (i) − (ii) | (iv) | (v) | (iv) − (v) |
| **Bayesian log. regr.** | | | | | | |
| a1a | -654.79 | -655.51 | 0.72 | -637.23 | -636.40 | -0.82 |
| australian | -268.36 | -269.35 | 0.99 | -256.82 | -256.73 | -0.09 |
| ionosphere | -138.30 | -139.62 | 1.31 | -124.44 | -124.35 | -0.08 |
| madelon | -2,466.28 | -2,466.15 | -0.13 | -2,600.32 | -2,399.65 | -200.67 |
| mushrooms | -210.00 | -211.43 | 1.42 | -180.60 | -179.89 | -0.72 |
| sonar | -149.58 | -151.69 | 2.11 | -110.33 | -110.04 | -0.29 |
| **Stan models** | | | | | | |
| congress | 421.91 | 421.79 | 0.12 | 423.58 | 423.55 | 0.04 |
| election88 | -1,419.02 | -1,420.01 | 0.99 | -1,645.18 | -1,398.03 | -247.15 |
| election88Exp | -1,376.03 | -1,380.18 | 4.15 | — | -1,381.79 | — |
| electric | -788.84 | -788.89 | 0.05 | -859.26 | -786.91 | -72.35 |
| electric-one-pred | -818.33 | -818.36 | 0.03 | -818.00 | -818.01 | 0.01 |
| hepatitis | -560.43 | -560.44 | 0.01 | -618.76 | -557.36 | -61.40 |
| hiv-chr | -608.42 | -608.77 | 0.35 | — | -582.78 | — |
| irt | -15,888.03 | -15,887.92 | -0.11 | -15,936.06 | -15,884.67 | -51.40 |
| mesquite | -30.08 | -30.15 | 0.08 | -29.78 | -29.83 | 0.05 |
| radon | -1,210.65 | -1,210.70 | 0.05 | -1,216.92 | -1,209.46 | -7.46 |
| wells | -2,042.37 | -2,042.45 | 0.08 | -2,041.90 | -2,041.95 | 0.05 |

Table 2: Comparison of Adam and SAA for VI: Median of the highest **ELBO** achieved across multiple optimization runs with different seeds for each model and approximating distribution. Adam was optimized using step-sizes of 0.1, 0.01, and 0.001, and the configuration with the highest median ELBO is reported. We additionally included the difference between the median ELBO achieved by Adam and SAA for VI: negative values indicate that SAA for VI achieved a higher ELBO than Adam. For further details, see Section 5.1. The full results are provided in Tables 6 and 7 in the appendix.

We conducted two comparisons for our study. First, we compared the median ELBO, obtained across 20 repetitions, at the end of the optimization process using Adam and SAA for VI. Initially, we ran the Adam experiments for $40,000$ iterations, but we found that for some models, there was a persistent large gap between the maximum median ELBO achieved with Adam and that of SAA for VI. We increased the maximum number of iterations to reduce the gap for models such as `election88`, `electric`, `irt`, `madelon`, and `radon`. (See Table 8 in the appendix). Table 2 presents the comparisons of median ELBOs. Although the Adam optimizer achieves a slightly higher median ELBO for some models—due to the stopping criterion of SAA for VI—SAA for VI achieves a noticeably higher median ELBO for complex models. We also observed that Adam diverged for models such as `election88Exp`. Additionally, Adam diverged for the `hepatitis` model when optimized for more than $40,000$ iterations, which partially explained the large gap between the median ELBOs of Adam and SAA for VI. We note that it's possible that Adam could achieve higher ELBO values by searching over a finer step-size grid; however, it is exactly this type of difficult and time-intensive tuning we seek to avoid with SAA.

Second, we compared the time taken to achieve a given ELBO. For each combination of model and approximating distribution, we computed the minimum between the median ELBO achieved by Adam and the median ELBO achieved by SAA for VI. This allows us to determine a value of the ELBO that was achieved for at least 50% of the runs, regardless of the optimization. To compare the performance of the algorithms fairly, we measured the time taken to reach an ELBO value within 1 nat of the determined minimum median ELBO across all runs. We computed this adjusted time for each run, ensuring the comparison is not influenced by our choices of maximum number of iterations for Adam.

Table 3 presents the time (in seconds) required to achieve the adjusted ELBO when using Adam and our proposed method, and the ratio between them. For example, running optimization for the `electric` model takes a minute when using Adam, as opposed to less than 2 seconds when using SAA for VI. In other words, Adam takes more than 30 times longer to achieve the adjusted ELBO as SAA for VI. It is worth noting that SAA for VI was at a disadvantage in the comparison, because the actual compute time required by Adam was three times larger than the reported one due to the selection of the step-size. As the evaluation of the model for different sample draws is vectorized on the GPU, the wall clock time in seconds serves as the most meaningful metric for comparing the compute time of both methods. Given the consistency of the results, we can confidently conclude that SAA for VI is a faster alternative to Adam in this case.

To explore the influence of sample size on Adam's performance, we provide a concise discussion in Appendix G, analyzing the results with sample sizes of 1 and 256. In the same appendix, we also incorporate Adagrad [Duchi et al., 2011] with a sample size of 16 as an alternative optimization method to Adam. Throughout our various experiments, SAA for VI consistently proved to be a robust alternative.

### 5.1.2 Batched quasi-Newton

As noted earlier in Section 4, our method exhibits certain differences compared to the batched quasi-Newton technique developed by Liu and Owen [2021], which also integrates second-order information into VI. In this section, we aim to empirically highlight the significance of these differences, specifically the use of a sequence of sample average approximations with an increasing number of samples.

To carry out this comparison, we implemented the batched quasi-Newton method in PyTorch without employing quasi-Monte Carlo sampling and compared it to our method. We ran the experiments for 40,000 iterations, with 20 independent runs for each. Initially, we used a sample size of 16 and increased it by a factor of 2 for models where the method encountered difficulties, up to a maximum of 128 samples. We consistently used $B = 20$ as recommended in the original paper.

When employing a simpler approximating distribution, such as a Gaussian distribution with a diagonal covariance matrix, the batched quasi-Newton method demonstrates performance on par with SAA for VI (refer to Table 9 in the appendix). However, the method encounters difficulties when using a more complex Gaussian distribution with a dense covariance matrix as the approximating distribution.

Table 4 displays the median final ELBO across runs for various models. The batched quasi-Newton method reaches optimal performance for most Bayesian logistic regression models but faces difficulties with models from the Stan example library. Even when increasing the sample size to 128, a significantly larger sample size than commonly employed with SGD, the method still falls short of reaching the optimal value. Additionally, we show in the appendix that the wall-clock time taken by the batched quasi-Newton method is often similar to or slower than the time taken by SAA for VI.

### 5.2 Ablation study

**Impact of warm start.** The optimization process requires a decision on whether to use warm start or draw fresh parameters for each iteration. Suppose that the inner optimization process Opt has already converged to parameters $\theta_t^*$. Despite the convergence, it may still be necessary to run the inner optimization process more times, as described in Section 3.2, to reduce overfitting. The question then arises whether it is computationally advantageous to use $\theta_t^*$ as the initial parameters or to draw a new set of parameters from a suitable distribution.

| | Diagonal Covariance | | | Dense Covariance | | |
|---|---|---|---|---|---|---|
| | Adam (i) | SAA for VI (ii) | Ratio (i)/(ii) | Adam (iv) | SAA for VI (v) | Ratio (iv)/(v) |
| **Bayesian log. regr.** | | | | | | |
| a1a | 18.09 | 0.38 | 48.24 | 19.95 | 19.69 | 1.01 |
| australian | 15.21 | 0.21 | 70.76 | 14.73 | 4.81 | 3.06 |
| ionosphere | 11.44 | 0.17 | 67.64 | 13.47 | 4.33 | 3.11 |
| madelon | 21.02 | 0.82 | 25.62 | 223.55 | 58.52 | 3.82 |
| mushrooms | 27.23 | 0.37 | 73.25 | 29.11 | 17.30 | 1.68 |
| sonar | 11.76 | 0.30 | 39.47 | 11.74 | 12.17 | 0.96 |
| **Stan models** | | | | | | |
| congress | 36.56 | 0.95 | 38.56 | 50.34 | 0.82 | 61.46 |
| election88 | 283.19 | 12.11 | 23.39 | 1,465.89 | 199.76 | 7.34 |
| election88Exp | 261.83 | 12.35 | 21.19 | — | 83.68 | — |
| electric | 65.14 | 1.92 | 33.96 | 235.40 | 42.14 | 5.59 |
| electric-one-pred | 55.22 | 0.51 | 107.75 | 70.62 | 0.62 | 114.40 |
| hepatitis | 103.89 | 2.74 | 37.88 | 264.52 | 96.09 | 2.75 |
| hiv-chr | 56.80 | 2.27 | 24.98 | — | 29.74 | — |
| irt | 33.53 | 1.70 | 19.67 | 210.05 | 94.80 | 2.22 |
| mesquite | 28.87 | 0.73 | 39.47 | 48.54 | 0.27 | 179.91 |
| radon | 74.83 | 1.57 | 47.72 | 252.85 | 18.66 | 13.55 |
| wells | 16.87 | 0.69 | 24.34 | 18.33 | 0.08 | 221.36 |

Table 3: Comparison of **running time**, in seconds, for Adam and SAA for VI across different datasets and distribution approximations, and Adam to SAA time ratio. Values of ratio greater than 1 indicate that Adam is slower than SAA for VI. SAA for VI generally outperforms Adam, with the exception of the `sonar` dataset. When using the diagonal covariance approximation, the speed improvement for SAA for VI is notably higher, reaching at least an order of magnitude in most cases. See Section 5.1 for more information.

Pasupathy [2010] provides an intuition of why using a warm start is helpful: in principle, the optimization process for larger sample sizes begin from a place that probably is close to a solution. However, we wanted to empirically verify this intuition. To determine the most efficient approach, we conducted an experiment to compare the performance of warm start and drawing fresh parameters across different models and approximating distributions. For each combination of models and distribution, we ran the sequence of SAA problems until convergence, using either warm start or by sampling new parameters at the beginning of each inner optimization. Specifically, for the sequence of sample sizes $(n_t)_{t\in\mathbb{N}}$ described above, we ran the inner optimization process Opt until it converged. At each iteration $t$, we initialized the process either with the previously computed optimal parameters $\theta_{t-1}^*$ (warm start) or by drawing a new random set of parameters (fresh start). We continue this process until the algorithm converges. We again used 20 repetitions for each configuration and report the median results. Our results, presented in Table 5, show that although the difference in nats between the median run is small, using warm start results in a significant reduction in the total time taken to converge. For example, on the `election88` dataset, using fresh samples takes 20× more time than using a warm start due to the inner optimization process Opt taking more iterations to find a good solution at each step.

| | | Dense Covariance | | | SAA for VI |
|---|---|---|---|---|---|
| | | Batched quasi-Newton—Sample Size | | | |
| | 16 | 32 | 64 | 128 | |
| **Bayesian log. regr.** | | | | | |
| a1a | -636.49 | | | | -636.40 |
| australian | -256.80 | | | | -256.73 |
| ionosphere | -124.44 | | | | -124.35 |
| madelon ✗ | -2,418.04 | -2,412.23 | -2,407.44 | -2,406.27 | -2,399.65 |
| mushrooms | -179.96 | | | | -179.89 |
| sonar | -110.09 | | | | -110.04 |
| **Stan models** | | | | | |
| congress | 423.59 | | | | 423.55 |
| election88 ✗ | $-1.15 \times 10^{12}$ | $-8.26 \times 10^{11}$ | $-7.23 \times 10^{11}$ | $-5.87 \times 10^{11}$ | -1,398.03 |
| election88Exp ✗ | $-3.47 \times 10^{19}$ | $-1.15 \times 10^{18}$ | $-3.72 \times 10^{16}$ | $-1.86 \times 10^{16}$ | -1,381.79 |
| electric ✗ | $-5.44 \times 10^{10}$ | $-6.20 \times 10^{9}$ | $-5.05 \times 10^{9}$ | $-6.08 \times 10^{9}$ | -786.91 |
| electric-one-pred | -1,145.79 | -818.00 | | | -818.01 |
| hepatitis ✗ | $-1.99 \times 10^{10}$ | $-1.03 \times 10^{10}$ | $-9.56 \times 10^{9}$ | $-1.64 \times 10^{10}$ | -557.36 |
| hiv-chr ✗ | $-6.44 \times 10^{15}$ | $-1.47 \times 10^{16}$ | $-3.59 \times 10^{15}$ | $-1.87 \times 10^{15}$ | -582.78 |
| irt ✗ | -20,481.68 | -18,573.30 | -17,263.15 | -16,099.44 | -15,884.67 |
| mesquite | -29.78 | | | | -29.83 |
| radon | $-1.58 \times 10^{6}$ | $-5.50 \times 10^{5}$ | -4,473.35 | -1,209.47 | -1,209.46 |
| wells | -2,041.90 | | | | -2,041.95 |

Table 4: **ELBO** achieved by the batched quasi-Newton method for VI using a Gaussian distribution with a dense covariance matrix, as proposed by Liu and Owen [2021]. The results for SAA for VI are included as a benchmark (refer to column (v) of Table 2). It is observed that the batched quasi-Newton method frequently converges to suboptimal solutions, indicated by ✗, especially in models from the Stan examples repository. In certain cases, such as the `election88` dataset, the SAA for VI method demonstrates a significant performance advantage over the batched quasi-Newton method. The initial sample size for the batched quasi-Newton method was set to 16 and increased when necessary to enhance the method's ELBO.

# 6 Conclusion

In this paper, we introduced the SAA for VI algorithm, which provides an effective and accurate solution to variational inference problems, significantly reducing the reliance on manual hyperparameter tuning. This promising method enhances both efficiency and precision in addressing these challenges.

| | (Fresh start) − (Warm start) ELBO difference | | (Fresh start)/(Warm start) Time ratio | |
|---|---|---|---|---|
| | Diagonal | Dense | Diagonal | Dense |
| **Bayesian log. regr.** | | | | |
| a1a | 0.00 | 0.00 | 1.11 | 1.78 |
| australian | 0.00 | 0.00 | 1.01 | 1.58 |
| ionosphere | 0.00 | 0.00 | 0.94 | 1.26 |
| madelon | 0.00 | 0.00 | 1.63 | 1.73 |
| mushrooms | 0.00 | 0.00 | 1.31 | 2.04 |
| sonar | 0.00 | 0.00 | 1.12 | 1.44 |
| **Stan models** | | | | |
| congress | 0.01 | 0.02 | 1.14 | 3.07 |
| election88 | -1.77 | 1.66 | 3.11 | 20.63 |
| election88Exp | -3.46 | 3.43 | 2.16 | 2.59 |
| electric | 0.00 | 0.00 | 2.64 | 4.69 |
| electric-one-pred | 0.01 | 0.00 | 1.05 | 0.75 |
| hepatitis | 0.01 | -0.04 | 2.77 | 2.03 |
| hiv-chr | 0.07 | -0.05 | 2.10 | 2.70 |
| irt | 0.00 | 0.00 | 3.63 | 6.56 |
| mesquite | 0.00 | 0.00 | 0.98 | 1.31 |
| radon | 0.00 | 0.00 | 2.29 | 5.35 |
| wells | 0.00 | 0.00 | 0.96 | 0.99 |

Table 5: Median **ELBO** variation in nats resulting from switching between two approaches: fresh start, where parameters are refreshed at each iteration to warm start, where previously learned parameters are used as the starting point. (Negative values indicate that the warm start approach is better.) We also provide the ratio of median **time taken** by the fresh start approach compared to the warm start approach. (Values larger than 1 indicate that the warm start approach is faster.) Our results indicate that warm start approaches can significantly reduce the optimization time required.

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

## A Detailed comparison with Adam

We now provide additional details about the experimental setup presented in Section 5.1. We used the Adam optimizer with the default parameters from the `torch.optim` package in PyTorch [Paszke et al., 2019], except for the step-size, which we varied across 0.1, 0.01, and 0.001. To approximate the distributions, we used a Gaussian with a diagonal covariance matrix and a more expressive Gaussian with a dense covariance matrix. Tables 6 and 7 present the experiment results disaggregated by step-size. In all cases we ran 20 repetitions of the experiments, and we estimated the objective function using 16 samples from the variational approximation $q_{\theta_t}$. Every 100 iterations we estimate the ELBO using $10,000$ fresh samples from $q_{\theta_t}$. Initially, we ran the experiments for $40,000$ iterations, but we found that the dense approximation produced unsatisfactory results for some models. We, therefore, increased the number of iterations for those models, but observed only slight changes in the maximum achieved ELBO, as shown in Table 8 and Table 7. It is also worth noting that the `hepatitis` model diverged when we ran it for more than $40,000$ iterations using the dense approximation.

| | Adam—Step Size | | | SAA for VI |
|---|---|---|---|---|
| | 0.1 | 0.01 | 0.001 | |
| **Bayesian log. regr.** | | | | |
| a1a | -656.19 | -654.98 | -654.79 | -655.51 |
| australian | -268.85 | -268.42 | -268.36 | -269.35 |
| ionosphere | -138.87 | -138.38 | -138.30 | -139.62 |
| madelon | -2,494.73 | -2,470.07 | -2,466.28 | -2,466.15 |
| mushrooms | -210.97 | -210.22 | -210.00 | -211.43 |
| sonar | -151.09 | -149.80 | -149.58 | -151.69 |
| **Stan models** | | | | |
| congress | 421.86 | 421.90 | 421.91 | 421.79 |
| election88 | -1,436.20 | -1,420.16 | -1,419.02 | -1,420.01 |
| election88Exp | -1,376.35 | -1,376.03 | -1,381.95 | -1,380.18 |
| electric | -790.66 | -789.06 | -788.84 | -788.89 |
| electric-one-pred | -818.34 | -818.33 | -1,063.98 | -818.36 |
| hepatitis | -564.05 | -560.83 | -560.43 | -560.44 |
| hiv-chr | -611.75 | -608.82 | -608.42 | -608.77 |
| irt | -15,896.00 | -15,889.39 | -15,888.03 | -15,887.92 |
| mesquite | -30.09 | -30.08 | -30.08 | -30.15 |
| radon | -1,211.57 | -1,210.79 | -1,210.65 | -1,210.70 |
| wells | -2,042.38 | -2,042.37 | -2,042.37 | -2,042.45 |

Table 6: Maximum **ELBO** achieved by Adam and SAA for VI with Gaussian distribution and **diagonal** covariance matrix as approximating distribution: median across seeds. The table shows the median of the maximum ELBO achieved by Adam and SAA for each model when using a Gaussian distribution with diagonal covariance matrix as approximating distribution. For each step-size used with Adam, we ran the algorithm 20 times and reported the median of the maximum ELBO achieved.

| | Adam—Step Sizes | | | SAA for VI |
|---|---|---|---|---|
| | 0.1 | 0.01 | 0.001 | |
| **Bayesian log. regr.** | | | | |
| a1a | -1,355.11 | -646.20 | -637.23 | -636.40 |
| australian | -269.97 | -257.53 | -256.82 | -256.73 |
| ionosphere | -148.71 | -125.21 | -124.44 | -124.35 |
| madelon | -66,648.98 | -7,599.58 | -2,600.32 | -2,399.65 |
| mushrooms | -242.99 | -182.65 | -180.60 | -179.89 |
| sonar | -386.12 | -114.58 | -110.33 | -110.04 |
| **Stan models** | | | | |
| congress | 423.36 | 423.53 | 423.58 | 423.55 |
| election88 | — | -1,645.18 | — | -1,398.03 |
| election88Exp | — | — | — | -1,381.79 |
| electric | — | -859.26 | — | -786.91 |
| electric-one-pred | -818.01 | -818.00 | -1,083.04 | -818.01 |
| hepatitis | — | -618.76 | — | -557.36 |
| hiv-chr | — | — | — | -582.78 |
| irt | -126,355.62 | -18,773.00 | -15,936.06 | -15,884.67 |
| mesquite | -29.80 | -29.79 | -29.78 | -29.83 |
| radon | — | -1,216.92 | -43,570.33 | -1,209.46 |
| wells | -2,041.91 | -2,041.90 | -2,041.90 | -2,041.95 |

Table 7: Maximum **ELBO** achieved by Adam and SAA for VI with Gaussian distribution and **dense** covariance matrix as approximating distribution: median across seeds. The table shows the median of the maximum ELBO achieved by Adam and SAA for each model when using a gaussian distribution with dense covariance matrix as approximating distribution. For each step-size used with Adam, we ran the algorithm 20 times and reported the median of the maximum ELBO achieved.

| | Adam | |
|---|---|---|
| | Diagonal Covariance | Dense Covariance |
| **Bayesian log. regr.** | | |
| a1a | 40,000 | 40,000 |
| australian | 40,000 | 40,000 |
| ionosphere | 40,000 | 40,000 |
| madelon | 40,000 | 400,000 |
| mushrooms | 40,000 | 40,000 |
| sonar | 40,000 | 40,000 |
| **Stan models** | | |
| congress | 40,000 | 40,000 |
| election88 | 40,000 | 400,000 |
| election88Exp | 40,000 | 40,000 |
| electric | 40,000 | 400,000 |
| electric-one-pred | 40,000 | 40,000 |
| hepatitis | 40,000 | 40,000 |
| hiv-chr | 40,000 | 40,000 |
| irt | 40,000 | 200,000 |
| mesquite | 40,000 | 40,000 |
| radon | 40,000 | 400,000 |
| wells | 40,000 | 40,000 |

Table 8: Maximum number of **iterations** for Adam optimization using Gaussian distribution with **diagonal** or **dense** covariance matrix. Some models (`election88`, `electric`, `irt`, `madelon`, and `radon`) were run for up to 10 times more iterations to achieve a comparable ELBO to SAA for VI.

# B   Detailed comparison with batched quasi-Newton

In this section, we provide further details about the experiments conducted using the batched quasi-Newton method of Liu and Owen [2021]. Table 9 compares the performance of batched quasi-Newton with our method when the approximating distribution is a Gaussian distribution with a diagonal covariance matrix. This table complements Table 4. As mentioned earlier, the results in this setting are quite similar to ours.

Additionally, we report the wall-clock time for each experiment in Table 10. We executed each experiment for 40,000 iterations and performed 20 independent runs for each one. Our method incorporates a stopping criterion based on convergence. To ensure a fair comparison with batched quasi-Newton, we need to detect when the algorithm converges. To approximate this, we first calculate the highest ELBO for each of the 20 independent runs using both batched quasi-Newton and SAA for VI. Then, we compute the median ELBO value across the repetitions for each method. Finally, we determine the minimum median ELBO value between the two methods and calculate the total time taken until the algorithm reaches within 1 nat of this minimum median ELBO value. These results are presented in Table 10.

Similar to the experiments with Adam, this calculation does not account for the time spent on sample sizes that were not useful.

|  | Diagonal Gaussian | |
| --- | --- | --- |
|  | Batched quasi-Newton 16 | SAA for VI |
| **Bayesian log. regr.** | | |
| a1a | -654.94 | -655.51 |
| australian | -268.47 | -269.35 |
| ionosphere | -138.49 | -139.62 |
| madelon | -2,466.58 | -2,466.15 |
| mushrooms | -210.26 | -211.43 |
| sonar | -150.14 | -151.69 |
| **Stan models** | | |
| congress | 421.91 | 421.79 |
| election88 | -1,426.01 | -1,420.01 |
| election88Exp | -1,382.64 | -1,380.18 |
| electric | -788.89 | -788.89 |
| electric-one-pred | -818.33 | -818.36 |
| hepatitis | -560.58 | -560.44 |
| hiv-chr | -608.58 | -608.77 |
| irt | -15,888.14 | -15,887.92 |
| mesquite | -30.08 | -30.15 |
| radon | -1,210.73 | -1,210.70 |
| wells | -2,042.37 | -2,042.45 |

Table 9: Comparison of the **ELBOs** obtained by batched quasi-Newton and SAA for VI when using a diagonal Gaussian distribution as the approximating distribution. The batched quasi-Newton method of Liu and Owen [2021] is executed using a sample size of 16. Median results are reported from 20 independent runs for each model. The corresponding results for SAA for VI can also be found in column (ii) of Table 2.

| | Diagonal Covariance | | | Dense Covariance | | |
|---|---|---|---|---|---|---|
| | Batched quasi-Newton | SAA for VI | Ratio | Batched quasi-Newton | SAA for VI | Ratio |
| | (i) | (ii) | (i)/(ii) | (iv) | (v) | (iv)/(v) |
| **Bayesian log. regr.** | | | | | | |
| a1a | 2.10 | 0.38 | 5.60 | 8.40 | 20.31 | 0.41 |
| australian | 1.08 | 0.21 | 5.03 | 2.55 | 4.81 | 0.53 |
| ionosphere | 1.10 | 0.17 | 6.50 | 2.35 | 4.33 | 0.54 |
| madelon | 7.82 | 0.81 | 9.71 | 384.02 | 62.98 | 6.10 |
| mushrooms ✗ | 2.26 | 0.37 | 6.07 | 7.31 | 18.84 | 0.39 |
| sonar | 1.28 | 0.30 | 4.28 | 3.72 | 12.48 | 0.30 |
| **Stan models** | | | | | | |
| congress | 2.93 | 0.95 | 3.08 | 4.99 | 0.82 | 6.10 |
| election88 ✗ | 1,660.06 | 8.96 | 185.34 | — | — | — |
| election88Exp ✗ | 799.40 | 9.75 | 82.02 | — | — | — |
| electric ✗ | 18.35 | 1.92 | 9.57 | — | — | — |
| electric-one-pred | 3.45 | 0.51 | 6.73 | 4.53 | 0.62 | 7.33 |
| hepatitis ✗ | 22.29 | 2.74 | 8.13 | — | — | — |
| hiv-chr ✗ | 30.57 | 2.27 | 13.44 | — | — | — |
| irt ✗ | 37.66 | 1.70 | 22.09 | 663.15 | 89.94 | 7.37 |
| mesquite | 1.39 | 0.73 | 1.90 | 0.95 | 0.27 | 3.51 |
| radon | 9.80 | 1.57 | 6.25 | 648.76 | 22.06 | 29.41 |
| wells | 1.04 | 0.69 | 1.49 | 0.50 | 0.08 | 6.08 |

Table 10: Comparison of **running times**, in seconds, to reach within 1 nat of the minimum median ELBO value between batched quasi-Newton and SAA for VI across different models and approximating distributions. Results for the approximation using a dense covariance matrix consider runs with a batched quasi-Newton sample size of 128. For models with ✗, indicating batched quasi-Newton failure in the dense covariance matrix approximation, only `madelon` and `irt` are reported, as they closely achieve the maximum ELBO. The table also presents the ratio of running times between the two methods; values greater than 1 indicate that SAA for VI is faster.

## C   Additional results for SAA for VI

Table 3 displays the median time taken by SAA for VI to reach the maximum ELBO attained by Adam. In this section, we present the total time taken by SAA for VI until completion. It is worth noting that, for certain models such as `election88`, SAA achieved an ELBO over 200 nats higher than Adam, which explain the differences between Table 11 and Table 3.

## D   Datasets description

We utilized the same datasets as Burroni et al. [2023]. The table below, adapted from their paper, provides a summary of the datasets employed.

| | Diagonal Covariance | | Dense Covariance | |
|---|---|---|---|---|
| | total time | maximum sample size | total time | maximum sample size |
| **Bayesian log. regr.** | | | | |
| a1a | 0.46 | $2^8$ | 52.99 | $2^{18}$ |
| australian | 0.22 | $2^6$ | 9.69 | $2^{17}$ |
| ionosphere | 0.16 | $2^6$ | 6.27 | $2^{16}$ |
| madelon | 1.11 | $2^{11}$ | 100.19 | $2^{18}$ |
| mushrooms | 0.42 | $2^8$ | 90.65 | $2^{17}$ |
| sonar | 0.29 | $2^8$ | 19.24 | $2^{18}$ |
| **Stan models** | | | | |
| congress | 0.95 | $2^5$ | 1.10 | $2^8$ |
| election88 | 12.84 | $2^8$ | 264.98 | $2^{15}$ |
| election88Exp | 11.65 | $2^{10}$ | 351.63 | $2^{12}$ |
| electric | 2.41 | $2^{11}$ | 70.07 | $2^{18}$ |
| electric-one-pred | 0.51 | $2^8$ | 0.62 | $2^7$ |
| hepatitis | 3.49 | $2^{12}$ | 163.19 | $2^{18}$ |
| hiv-chr | 2.68 | $2^9$ | 64.87 | $2^{18}$ |
| irt | 13.83 | $2^{14}$ | 473.77 | $2^{18}$ |
| mesquite | 0.73 | $2^5$ | 0.38 | $2^6$ |
| radon | 2.08 | $2^{11}$ | 53.62 | $2^{18}$ |
| wells | 0.70 | $2^5$ | 0.09 | $2^5$ |

Table 11: Median **running time** (in seconds) and corresponding median **sample size** at which convergence occurs for SAA for VI across runs. As described in Section 5, the sample size is limited to a maximum of $2^{18}$, which proved sufficient for all models.

Table 12: Description of datasets/models.

| | Num. of variables | Num. of records | Comments |
|---|---|---|---|
| **Bayesian log. regr.** | | | |
| a1a | 105 | 1605 | First 1605 instances of the Adult Data Set, following LIBSVM Chang and Lin [2011], + discretized continous and dummified. |
| australian | 35 | 690 | From UCI + dummified. |
| ionosphere | 35 | 351 | From UCI |
| madelon | 500 | 4400 | From UCI |
| mushrooms | 96 | 8124 | From UCI + dummified. |
| sonar | 61 | 208 | From UCI |
| **Stan models** | | | |
| congress | 4 | 343 | Gelman and Hill [2006] Ch. 7 |
| election88 | 95 | 2015 | Gelman and Hill [2006] Ch. 19 |
| election88Exp | 96 | 2015 | Gelman and Hill [2006] Ch. 19 |
| electric | 100 | 192 | Gelman and Hill [2006] Ch. 23 |
| electric-one-pred | 3 | 192 | Gelman and Hill [2006] Ch. 23 |
| hepatitis | 218 | 288 | WinBUGS Lunn et al. [2000] examples |
| hiv-chr | 173 | 369 | Gelman and Hill [2006] Ch. 7 |
| irt | 501 | 30105 | Gelman and Hill [2006] Ch. 14 |
| mesquite | 3 | 46 | Gelman and Hill [2006] Ch. 4 |
| radon | 88 | 919 | `radon-chr` from Gelman and Hill [2006] Ch. 19 |
| wells | 2 | 3020 | Gelman and Hill [2006] Ch. 7 |

## E   Addendum

As mentioned in the related work section, a result by Giordano et al. [2023] demonstrates the futility of using a sample size smaller than the dimension of the latent space for the ELBO optimization problem. In this section, we provide a proof sketch of this result, adapted to our notation.

**Theorem E.1** (Theorem 2 of Giordano et al. [2023])**.** *Let $q_\theta$ be a Gaussian distribution with parameters $\theta = (\mu, LL^{\mathrm{T}})$, where $\mu \in \mathbb{R}^d$ and $L \in \mathbb{R}^{d \times d}$ is a lower-triangular matrix with positive diagonal elements. If we draw a sample of size $n < d$ from $q_{\mathrm{base}}$, denoted by $\boldsymbol{\epsilon} = \epsilon_1, \ldots, \epsilon_n$, then the optimization problem in Eq.* (5) *is unbounded:*

$$\sup_{\theta \in \Theta} \hat{\mathcal{L}}_{\boldsymbol{\epsilon}}(\theta) = \sup_{\theta \in \Theta} \frac{1}{n} \sum_{i=1}^{n} [\ln p(z_\theta(\epsilon_i), x) - \ln q_\theta(z_\theta(\epsilon_i))] = \infty.$$

*Proof.* Since $n < d$, there exists a nonzero vector $\mathbf{v} \in \mathbb{R}^d$ such that $\langle \mathbf{v}, \epsilon_i \rangle = 0$ for all $1 \leq i \leq n$. Without loss of generality, assume that the largest index $\ell$ with $\mathbf{v}_\ell \neq 0$ satisfies $\mathbf{v}_\ell = 1$. Define the lower triangular matrix

$$L_\lambda = \begin{pmatrix} I_{\ell-1} & & \mathbf{0} \\ & \lambda \mathbf{v}^{\mathrm{T}} & \\ \mathbf{0} & & I_{d-\ell.} \end{pmatrix}.$$

Then, we have $(L_\lambda \epsilon_i)_\ell = 0 = (L_0 \epsilon_i)_\ell$ for all $1 \leq i \leq n$. Let $\theta_\lambda = (\mathbf{0}, L_\lambda L_\lambda^{\mathrm{T}})$. For $\lambda > 0$, we obtain

$$\hat{\mathcal{L}}_{\boldsymbol{\epsilon}}(\mathbf{0}, L_\lambda L_\lambda^{\mathrm{T}}) = \frac{1}{n} \sum_{i=1}^{n} [\ln p(L_\lambda \epsilon_i, x) - \ln q_{\theta_\lambda}(L_\lambda \epsilon_i)] = \frac{1}{n} \sum_{i=1}^{n} [\ln p(L_0 \epsilon_i, x) - \ln q_{\theta_\lambda}(L_0 \epsilon_i)] = c + \ln \lambda,$$

where $c$ is a constant independent of $\lambda$.

The result follows by letting $\lambda \to \infty$. □

With this result in mind, we decided to adapt the SAA for VI algorithm by, in the case of a dense covariance matrix approximation, drawing a sample of size $n$, set as the smallest power of two exceeding twice the latent space dimension. Table 13 and 14 present the experimental results alongside the previously computed results. As observed, starting with a larger sample size allows us to reduce the number of iterations required to achieve a certain accuracy. Furthermore, this reduction is substantial. This outcome was anticipated because, when the problem was unbounded, the optimization process for smaller $n$ typically concluded when the maximum number of iterations was reached, meaning the entire computational budget was utilized.

| | Adam | SAA for VI original, min $n = 32$ | | | SAA for VI new, min $n > d$ | | |
|---|---|---|---|---|---|---|---|
| | Time (i) | Min $n$ | Time (ii) | Ratio (i)/(ii) | Min $n$ | Time (iii) | Ratio (i)/(iii) |
| **Bayesian log. regr.** | | | | | | | |
| a1a | 19.95 | 32 | 19.69 | 1.01 | 256 | 4.69 | 4.26 |
| australian | 14.73 | 32 | 4.81 | 3.06 | 128 | 1.14 | 12.96 |
| ionosphere | 13.47 | 32 | 4.33 | 3.11 | 128 | 0.80 | 16.85 |
| madelon | 223.55 | 32 | 58.52 | 3.82 | 1,024 | 2.57 | 86.90 |
| mushrooms | 29.11 | 32 | 17.30 | 1.68 | 256 | 4.43 | 6.57 |
| sonar | 11.74 | 32 | 12.17 | 0.96 | 128 | 2.75 | 4.27 |
| **Stan models** | | | | | | | |
| congress | 50.34 | 32 | 0.82 | 61.46 | 32 | 0.78 | 64.40 |
| election88 | 1,465.89 | 32 | 199.76 | 7.34 | 256 | 45.72 | 32.06 |
| election88Exp | — | 32 | 83.68 | — | 256 | 5.59 | — |
| electric | 235.40 | 32 | 42.14 | 5.59 | 256 | 13.27 | 17.74 |
| electric-one-pred | 70.62 | 32 | 0.62 | 114.40 | 32 | 0.60 | 117.46 |
| hepatitis | 264.52 | 32 | 96.09 | 2.75 | 512 | 11.49 | 23.02 |
| hiv-chr | — | 32 | 29.74 | — | 512 | 4.11 | — |
| irt | 210.05 | 32 | 94.80 | 2.22 | 1,024 | 15.38 | 13.65 |
| mesquite | 48.54 | 32 | 0.27 | 179.91 | 32 | 0.26 | 185.76 |
| radon | 252.85 | 32 | 18.66 | 13.55 | 256 | 7.43 | 34.03 |
| wells | 18.33 | 32 | 0.08 | 221.36 | 32 | 0.08 | 232.47 |

Table 13: Comparison of **running time**, in seconds, for Adam and SAA for VI across various datasets, using a Gaussian approximating distribution with a dense covariance matrix and calculating the Adam to SAA time ratio. The **minimum sample size** $n$ for SAA in VI is also displayed. We consider two settings: one where the minimum $n$ is set to 32 for all datasets, which corresponds to the configuration used in this paper [cf. Table 3], and another where the minimum sample size is chosen as the nearest power of 2 to twice the number of parameters $d$ in the model. The results indicate that by avoiding the use of small sample sizes, the running time of SAA in VI can be significantly reduced.

| | Batched quasi-Newton | SAA for VI original, min $n = 32$ | | | SAA for VI new, min $n > d$ | | |
|---|---|---|---|---|---|---|---|
| | Time (i) | Min $n$ | Time (ii) | Ratio (i)/(ii) | Min $n$ | Time (iii) | Ratio (i)/(iii) |
| **Bayesian log. regr.** | | | | | | | |
| a1a | 8.40 | 32 | 20.31 | 0.41 | 256 | 5.32 | 1.58 |
| australian | 2.55 | 32 | 4.81 | 0.53 | 128 | 1.14 | 2.24 |
| ionosphere | 2.35 | 32 | 4.33 | 0.54 | 128 | 0.80 | 2.93 |
| madelon | 384.02 | 32 | 62.98 | 6.10 | 1,024 | 7.22 | 53.22 |
| mushrooms ✗ | 7.31 | 32 | 18.84 | 0.39 | 256 | 5.94 | 1.23 |
| sonar | 3.72 | 32 | 12.48 | 0.30 | 128 | 2.95 | 1.26 |
| **Stan models** | | | | | | | |
| congress | 4.99 | 32 | 0.82 | 6.10 | 32 | 0.78 | 6.39 |
| election88 ✗ | | | | | | | |
| election88Exp ✗ | | | | | | | |
| electric ✗ | | | | | | | |
| electric-one-pred | 4.53 | 32 | 0.62 | 7.33 | 32 | 0.60 | 7.53 |
| hepatitis ✗ | | | | | | | |
| hiv-chr ✗ | | | | | | | |
| irt ✗ | 663.15 | 32 | 89.94 | 7.37 | 1,024 | 7.24 | 91.55 |
| mesquite | 0.95 | 32 | 0.27 | 3.51 | 32 | 0.26 | 3.63 |
| radon | 648.76 | 32 | 22.06 | 29.41 | 256 | 10.67 | 60.78 |
| wells | 0.50 | 32 | 0.08 | 6.08 | 32 | 0.08 | 6.38 |

Table 14: Comparison of **running time** (in seconds) between batched quasi-Newton and SAA for VI on various datasets, using a Gaussian approximating distribution with a dense covariance matrix and calculating the batched quasi-Newton to SAA time ratio. The **minimum sample size** $n$ for SAA in VI is displayed. For models where the batched quasi-Newton method did not fully converge (✗), we only show results for mushrooms and irt, as the others diverged. Two settings are considered: one with a minimum $n$ of 32 for all datasets (used in this paper [cf. Table 10]), and another with the minimum sample size set to the nearest power of 2 greater than twice the number of parameters $d$ in the model. As in Table 13, the results indicate that avoiding small sample sizes can significantly reduce the running time of SAA in VI.

## F  Hyperparameters

As with any optimization algorithm, our implementation of the SAA for VI algorithm uses certain constants and hyperparameters. Table 15 details the purpose of each such number, along with the rationale behind our chosen values. We emphasize that SAA for VI performs well across many models without tuning these parameters (our experiments used a single setting): many can be considered constants, while others control tradeoffs between computation and precision in a straightforward way, such as tolerance parameters. While the current hyperparameter values are not tuned, we are open to the possibility of further enhancing the algorithm's performance through careful tuning.

The sequence of sample sizes is controlled by the first two hyperparameters. We tested a variety of exponentially increasing sequences and determined that the performance was largely unaffected by the specific choice. However, the initial sample size showed a more pronounced effect on performance as it could potentially 'save work' by avoiding smaller sample sizes if larger ones are required. This is not always predictable; our addendum, following Giordano et al. [2023]'s concurrent work, refines SAA for VI by tuning this value based on the model and approximation family.

The remaining hyperparameters, listed last in the table, mainly dictate when to halt the process. For example, a user may deem being 1 nat away from the optimum as adequate, thus setting $\delta$ to 1 instead of 0.01. The $\alpha$ (significance level for t-test) could also be adjusted depending on the desired balance between computation cost and approximation precision. Similar parameters are used in most implementations of other optimization algorithms (maximum iterations, absolute/relative tolerance, etc.) and tend to be less critical than parameters like step sizes as they affect the trade-off between computational time and numerical precision rather than the fundamental operation of the algorithm.

| Hyperparameter | Value | Purpose | Justification |
|---|---|---|---|
| Initial sample size ($n$) | 32 | Sets the starting point for the sample size sequence | Arbitrary choice. Refined based on the work of Giordano et al. [2023] in the addendum. |
| Sample size and max iterations sequence ($2n$, $2\tau$) | 2 | Determines progression of sample sizes and max inner optimizer iterations | Arbitrary. We tested alternative sequences with negligible performance impact |
| ELBO difference threshold ($\delta$) | 0.01 | Convergence criterion for the optimizer | Conservative choice ensuring precision |
| Max. number of SAA steps (max_t) or max. sample size | $2^{18}$ | Limits total number of SAA steps or sample size | Chosen to ensure optimization usually concludes for other reasons |
| Inner optimizer early exit count (`count` < 3) | 3 | Specifies how many times inner optimizer can finish after few iterations | We found empirically that this counter was necessary, but we didn't explore other alternatives. |
| `VERY_SMALL_ITER` for inner optimizer | 5 | Defines what is considered a small number of iterations for the inner optimizer | Arbitrary choice. It is related to the early exit `count`. |
| Significance level ($\alpha$) for t-test | 1% | Statistical significance criterion | Standard value in significance testing |
| Test set sample size | 10k | Size of the sample set for ELBO estimation | Arbitrary. It is related to $\alpha$ |
| Initial maximum number of iterations for inner optimizer ($\tau$) | 300 | Sets an initial limit for optimizer iterations | Arbitrary. However, it self-adjusts as needed |

Table 15: Hyperparameter choices for our SAA for VI experiments

# G   Additional experiments with Adam and AdaGrad

This section provides supplementary experimental findings with Adam and AdaGrad. We further explore the performance of Adam with two additional sample sizes: 1 and 256. For AdaGrad, we maintain the sample sizes consistent with those discussed in the main body of the text. The step-size search across 0.1, 0.01, and 0.001 remains unchanged in all experiments. As the sample size increases, the maximum ELBO value in most models tends towards the one obtained using SAA for VI, as demonstrated in Table 16. (We do not show the results for Adam with $n = 1$ because those were of poorer quality than the results for $n = 16$.) Despite this improvement, some models still exhibit significant disparity. It is important to note that Adam's computational cost continues to be higher than SAA for VI, as evidenced by Table 17. Note that the same instances of SAA for VI were used in all scenarios. Meaning, for each SAA for VI iteration, we ran Adam nine times. This repetition is not reflected in the presented numbers.

In AdaGrad's case, as shown in the Tables, there are promising results for Bayesian logistic regression models. However, the same performance does not extend to the Stan models. Only in the `wells` model does the maximum ELBO value closely match that of SAA for VI.

| | AdaGrad | Adam | SAA for VI | Differences | |
| --- | --- | --- | --- | --- | --- |
| | (n=16) | (n=256) | | AdaGrad - SAA | Adam - SAA |
| | (i) | (ii) | (iii) | (i) − (iii) | (ii) − (iii) |
| **Bayesian log. regr.** | | | | | |
| a1a | -636.76 | -636.57 | -636.40 | -0.36 | -0.17 |
| australian | -256.77 | -256.75 | -256.73 | -0.04 | -0.01 |
| ionosphere | -124.39 | -124.36 | -124.35 | -0.04 | -0.00 |
| madelon | -2,469.59 | -2,433.07 | -2,399.65 | -69.94 | -33.42 |
| mushrooms | -181.48 | -180.02 | -179.89 | -1.59 | -0.13 |
| sonar | -110.19 | -110.09 | -110.04 | -0.14 | -0.05 |
| **Stan models** | | | | | |
| congress | 413.88 | 423.59 | 423.55 | -9.66 | 0.05 |
| election88 | — | -1,446.37 | -1,398.03 | — | -48.34 |
| election88Exp | — | — | -1,381.79 | — | — |
| electric | — | -792.28 | -786.91 | — | -5.38 |
| electric-one-pred | -5,572.18 | -818.00 | -818.01 | -4,754.17 | 0.01 |
| hepatitis | — | -566.51 | -557.36 | — | -9.14 |
| hiv-chr | — | -77,190.31 | -582.78 | — | -76,607.53 |
| irt | -15,900.00 | -15,894.76 | -15,884.67 | -15.33 | -10.09 |
| mesquite | -75.93 | -29.78 | -29.83 | -46.10 | 0.05 |
| radon | — | -1,210.36 | -1,209.46 | — | -0.90 |
| wells | -2,041.90 | -2,041.90 | -2,041.95 | 0.05 | 0.05 |

Table 16: Comparison of AdaGrad and Adam to SAA for VI: Median of the highest **ELBO**.

| | AdaGrad | Adam | SAA for VI | Ratios | |
|---|---|---|---|---|---|
| | (n=16) | (n=256) | | AdaGrad/SAA | Adam/SAA |
| | (i) | (ii) | (iii) | (i)/(iii) | (ii)/(iii) |
| **Bayesian log. regr.** | | | | | |
| a1a | 12.67 | 16.16 | 20.31 | 0.62 | 0.80 |
| australian | 4.74 | 12.94 | 4.81 | 0.99 | 2.69 |
| ionosphere | 2.95 | 12.41 | 4.33 | 0.68 | 2.87 |
| madelon | 85.49 | 17.03 | 60.31 | 1.44 | 0.28 |
| mushrooms | 68.22 | 31.49 | 18.84 | 3.94 | 1.67 |
| sonar | 3.73 | 9.81 | 12.48 | 0.31 | 0.79 |
| **Stan models** | | | | | |
| congress | 18.36 | 39.40 | 0.82 | 22.41 | 48.11 |
| election88 | — | 3,485.68 | 200.34 | — | 17.40 |
| election88Exp | — | — | 83.68 | — | — |
| electric | — | 275.95 | 49.16 | — | 5.61 |
| electric-one-pred | 32.53 | 65.63 | 0.62 | 52.70 | 106.32 |
| hepatitis | — | 250.90 | 98.69 | — | 2.54 |
| hiv-chr | — | 157.79 | 29.91 | — | 5.28 |
| irt | 106.84 | 62.90 | 109.17 | 0.98 | 0.58 |
| mesquite | 39.74 | 39.76 | 0.27 | 147.28 | 147.35 |
| radon | — | 197.21 | 21.19 | — | 9.31 |
| wells | 7.18 | 17.48 | 0.08 | 86.66 | 211.03 |

Table 17: Comparison of **running time**, in seconds, for AdaGrad and Adam to SAA for VI.

