# OpenReview forum: "Sample Average Approximation for Black-Box Variational Inference"
_TMLR — Rejected by TMLR_

### Review · Reviewer_qePA · 2023-05-28

**Summary Of Contributions:**

This paper proposes a method for black-box variational inference that automates the hyperparameter selection in the problem, such as the number of Monte Carlo samples. The algorithm equates to solving a sequence of sample average approximation problems using quasi-Newton methods and line search. Experiments are done on datasets from the UCI dataset repository and Stan examples. Results show that the algorithm has competitive performance and less computational burden compared to Adam + grid search and improved performance compared to Liu and Owen (2021), which also uses quasi-Newton methods and line search.

**Audience:**

Yes

**Broader Impact Concerns:**

This paper discusses a general optimization approach, so I don't think that there are any ethical implications that need to be addressed. No broader impact statement was included.

**Claims And Evidence:**

No

**Requested Changes:**

## Critical ##
- The proposed algorithm still has some hyperparameters, such as the number of optimizer iterations and the significance level of the t-test. Please discuss how they were chosen. Why are they a better option than, say, the number of Monte Carlo samples? Is the performance more robust to changes in them?
- Please include some discussion of AutoML methods in the literature review and experimental comparison to one of them if possible.

## Helpful ##
- Recently black box VI has been widely applied in deep learning tasks, for example to provide uncertainty estimates. However, the datasets in the paper have dimension on the order of $10^2$. It would strengthen the work to provide results on a small neural network, such as LeNet for MNIST digit recognition.

**Strengths And Weaknesses:**

## Strengths ##
- The paper is well written and the proposed algorithm is clearly presented.
- The experimental protocol is sound. Comparisons to baselines are made based on both ELBO and computational burden, and ablations are included.

## Weaknesses ##
- The proposed algorithm also has some hyperparameters, but their selection was not discussed in many cases.
- No experiments are done on high dimensional problems such as neural networks, the highest dataset dimension is $~500$.
- The proposed algorithm tackles a similar problem to autoML, i.e. automatically learning hyperparameters, which has seen a lot of work in recent years, see the survey [1]. However, this connection is not mentioned and the experiments do not compare to one such method.

[1] He, Xin, Kaiyong Zhao, and Xiaowen Chu. "AutoML: A survey of the state-of-the-art." Knowledge-Based Systems 212 (2021): 106622.

---

> ### Author Response · Authors · 2023-06-07
> **Response to review**
>
> > The proposed algorithm still has some hyperparameters...
>
> This is a fair comment.
>  We added a table of all constants and hyperparameters in the SAA for VI algorithm to Appendix F that describes their purpose and how they were chosen. We also include some discussion about why performance is insensitive to these choices.
> Briefly, we expect a user will never need to change most of these values, and others govern things like stopping conditions, which control straightforward tradeoffs between computation and precision and are similar to tolerance parameters present in the implementations of most optimization algorithms.
>
> Our experiments show that SAA for VI performs robustly with one setting of hyperparameters.
> It’s useful to first highlight the hyperparameters that are not present in SAA for VI. The key hyperparameters for algorithms like Adam and AdaGrad are the step size and number of samples. Performance is highly sensitive to these parameters, which directly influence the sequence $\theta_t$ of approximate solutions. As demonstrated in our experiments (and supported by existing literature), searching over the step-size for these methods is crucial for good results, and the same applies to the sample size. A key aspect of SAA for VI is that one can view it as “tuning” both parameters internally: the line search determines the step size, and an increasing sequence of sample sizes with convergence criteria is used to select the sample size. The first two hyperparameters in the table for SAA for VI control the sequence of sample sizes. We tested several different exponentially increasing sequences and found performance was insensitive to the specific choice. Performance is more sensitive to the starting sample size as it may be possible to “save work” by skipping small sample sizes if a large one is needed, but it is not always clear in advance when this is the case; our addendum based on the concurrent work of Giordano et al. improves SAA for VI by tuning this value based on the model and approximating family.
>
> The remaining hyperparameters (the last seven in the table) primarily govern stopping. For instance, a user might consider being 1 nat away from the optimum sufficient, setting $\delta$ to 1 instead of 0.01. The $\alpha$ (significance level for t-test) could also be adjusted depending on the desired tradeoff between computational expense and precision of the approximation. Similar parameters are used in most implementations of other optimization algorithms (maximum iterations, absolute tolerance, relative tolerance, etc.) even if they are not considered part of the main algorithm. They tend to be less critical than parameters like step sizes as they affect the trade-off between computational time and numerical precision rather than the fundamental operation of the algorithm.
>
> > Please include some discussion of AutoML methods
>
> We added a brief mention of AutoML in our related work section. Despite some broad similarities, our work differs significantly in scope and intent from AutoML methods. Although there is a shared desire to minimize user intervention, AutoML is very broad in scope and focuses on all steps of constructing and deploying a predictive model, usually trained by loss minimization. In contrast, we are not training a model; there is a well defined model and probabilistic inference task, and we seek to approximate the posterior distribution.

---

> > ### Comment · Reviewer_qePA · 2023-06-16
> > **Thank you for your response**
> >
> > After reading your explanation on the algorithm's hyperparameters, I feel that my questions on that front have been adequately addressed. I agree that AutoML is a broader topic, I respectfully disagree that it is not relevant here. The posterior distribution is still a model of the data that will be used for downstream predictive tasks.

---

### Review · Reviewer_269H · 2023-05-30

**Summary Of Contributions:**

The script proposed a new method for solving the ELBO maximization problem. They first transform the stochastic objective to a deterministic one, then solve the problem using 2nd-order methods and line-search. Small-scale experiments are conducted to show the efficacy of the proposed method.

**Audience:**

Yes

**Broader Impact Concerns:**

Null

**Claims And Evidence:**

Yes

**Requested Changes:**

**Major issues**

1. The experiments are mostly conducted on toy datasets such as UCI. It is unclear whether the proposed method can work well in  more practical scenarios.



2. The proposed algorithm is mainly based on line-search and  2nd-order methods like LBFGS, which are both computationally expensive.  However, there is no discussion on the computational overhead.  We do not have evidence that the proposed methods can  be scaled-up to larger-sized experiments.





3. It is unfair to compare Adam with second-over methods like LBFGS. Adam is a first-order algorithm which is designed for large-scale experiments where Hessian is unavailable, so it is not surprising LBFGS outperforms Adam on small-scaled experiments since LBFGS uses more 2nd-order information.


**Other questions:**

1. I am a bit confused by the name "black-box VI": why do you call it black-box when you can access the gradient information? Usually, the term "black-box" is used when you only have access to the function value, but not the gradient information.


2. Please discuss the importance of the ELBO maximization problem in the era of large AI models.






**Strengths And Weaknesses:**

The presentation is mostly clear

---

> ### Author Response · Authors · 2023-06-07
> **Response to review**
>
> > The experiments are mostly conducted on toy datasets such as UCI
>
> See the [general response](https://openreview.net/forum?id=Lvg10LZ5nL&noteId=nJqrGbrF5K) and comments to [reviewer D3CN](https://openreview.net/forum?id=Lvg10LZ5nL&noteId=MkeJitysfl): these are challenging inference problems representative of real applied statistical modeling.
>
> > It is unfair to compare Adam with second-over methods like LBFGS. Adam is a first-order algorithm which is designed for large-scale experiments where Hessian is unavailable, so it is not surprising LBFGS outperforms Adam on small-scaled experiments since LBFGS uses more 2nd-order information.
>
> People have been working on black-box VI for around 10 years, and Adam is currently considered the accepted best practice. The intuition about that LBFGS might perform better is indeed a key motivation for our method. But there are two things that help understand its significance. First, “small-scale” models (up to hundreds of latent variables) are indeed of interest to many researchers, and probabilistic inference on such models is far from being solved. Second, it's not straightforward to apply LBFGS methods with line search in a stochastic optimization setting. One potential approach could be employing the batched quasi-Newton method proposed by Liu and Owen [2021]. We contributed an alternative and demonstrated in our experiments that it performs much better for complex models; it is also conceptually simpler.
>
> About computational overhead, we wish to point out that we included a running time comparison in all our experiments. Furthermore, we developed a protocol designed to make our experiments as informative as possible. We believe this approach gives a comprehensive picture of the performance of our method, including its computational efficiency.
>
> > I am a bit confused by the name "black-box VI"
>
> The term "black-box VI" likely stems from the title of the influential 2014 paper by Ranganath, Gerrish, and Blei, although related concepts had already been discussed in the literature, as we briefly touched upon in our paper's introduction. The designation "black-box" is employed in this context to differentiate from variational inference methods along the lines of those presented by Jaakkola and Jordan, as in their 1996 paper "A variational approach to Bayesian logistic regression models and their extensions". The term emphasizes the general applicability of these methods to a broad range of models without needing to explicitly derive model-specific updates, and is widely used in this way.

---

> > ### Comment · Reviewer_269H · 2023-06-17
> > **Thanks for the rebuttal**
> >
> > Thanks for the rebuttal.
> >
> > I still lack confidence whether the proposed method can be efficiently applied in large-scale practical settings.
> > In the rebuttal, the authors only emphasize that the small-scaled UCI tasks are nontrivial, but they did not address the above concern.

---

### Review · Reviewer_D3CN · 2023-05-31

**Summary Of Contributions:**

The paper introduces a method for black-box variational inference (VI) using sample average approximation (SAA). The authors aim to improve the accuracy and efficiency of variational inference methods by incorporating sample average approximation (SAA) and quasi-Newton methods, such as L-BFGS. The algorithm aims to maximize the evidence lower bound (ELBO) while balancing computational cost and overfitting. It incorporates second-order information and seamlessly integrates sample size consideration. Experimental evaluations compare the proposed method with Adam and batched quasi-Newton on various UCI datasets.

The key contributions are as follows:

+ Algorithm for Black-Box Variational Inference: The authors propose an algorithm based on SAA to approximate the solution of maximizing the evidence lower bound (ELBO) in black-box VI. They provide a detailed description of the algorithm, which aims to strike a balance between computational cost and overfitting.
+ Experimental Evaluations: The authors compare their method with two approaches; Adam and batched quasi-Newton. The later one incorporate second-order information. The authors conduct experiments to evaluate the performance of their proposed method. They compare it against Adam with a fixed step-size and batched quasi-Newton on various UCI datasets. The experiments aim to assess the performance of the proposed method in terms of optimization accuracy and computational efficiency.

**Audience:**

Yes

**Broader Impact Concerns:**

N/A.

**Claims And Evidence:**

Yes

**Requested Changes:**

See weaknesses for the details.

**Strengths And Weaknesses:**

Strengths:

+ Algorithm: The paper proposes a novel algorithm that combines SAA with quasi-Newton methods, specifically the well-known L-BFGS algorithm. This integration of SAA and L-BFGS is a strength, as it seems to improve the accuracy and efficiency of black-box VI.
+ Addressing Overfitting: The paper addresses the challenge of overfitting in VI by introducing a stopping criteria based on overfitting.
+ Convergence: The proposed algorithm incorporates heuristics for selecting sample sizes and determining when to halt the optimization process. This contributes to improved convergence and helps in finding better solutions to the stochastic optimization problem.

Weaknesses:

+ Lack of Comparative Analysis: While the paper presents experimental results comparing the proposed method with Adam and batched quasi-Newton, a more comprehensive comparative analysis could strengthen the evaluation. Including additional baselines or state-of-the-art methods would provide a clearer understanding of the strengths and limitations of the proposed algorithm. They mention AdaGrad, then I think they should include Adagrad in their experiments. Moreover, they should compare to (tuned) mini-batch Adam and AdaGrad, not only tuning the learning rate $\alpha$.
+ Theoretical Analysis: The paper lacks a thorough theoretical analysis of the proposed algorithm. While empirical results are provided, a deeper theoretical understanding of the algorithm's convergence properties, sample size selection, and stopping criteria would enhance the paper's contribution.
Recently, Godichon-Baggioni et al. (2022, 2023) showed how time-varying sample sizes (mini-batches) influence the convergence of stochastic gradient-based methods. In particular, they show how to tune the learning rate for an increasing sequence of sample sizes. Moreover, they show com this can be combined with Polyak-Ruppert averaging (or a weighted version) to accelerate convergence, or even break dependence. They authors write this averaging accelerate convergence...
+ Limited Scope of Applications: The experiments focus on a specific set of models with UCI datasets. A wider range of application domains and models would demonstrate the generalizability and effectiveness of the proposed method in various contexts.

References:
+ Godichon-Baggioni, A., Werge, N., & Wintenberger, O. (2023). Non-Asymptotic Analysis of Stochastic Approximation Algorithms for Streaming Data. ESAIM: PS 27 482-514.
+ Godichon-Baggioni, A., Werge, N., & Wintenberger, O. (2022). Learning from time-dependent streaming data with online stochastic algorithms. arXiv preprint arXiv:2205.12549.

---

> ### Author Response · Authors · 2023-06-07
> **Response to review (1/n)**
>
> > Lack of Comparative Analysis: [...]  they should compare to (tuned) mini-batch Adam and AdaGrad,
>
> Following your suggestion, we have run experiments that include Adam with sample sizes of 1 and 256, which adds to our original sample size of 16. Employing a larger sample size of 256 yielded a higher median ELBO at the end of the optimization compared to a sample size of 16. However, certain models still diverged or fell short of the SAA for VI results. Furthermore, despite improvements in ELBO, Adam’s computation time remained higher than SAA for VI.
>
> We also compared to AdaGrad with the same three learning rates and a sample size of 16. AdaGrad showed promising results for the Bayesian logistic regression models but the performance was poor for the Stan models: its maximum ELBO value was comparable to SAA for VI only for one model (‘wells’).
>
> In total, we compared SAA for VI to 12 different parameter settings of Adam and AdaGrad. No setting was universally best, and SAA for VI compared favorably to all different settings. Moreover, we always selected the best step size in hindsight for the comparison methods, without measuring the extra running time for tuning. In practice, a user would need to spend extra computational effort tuning the parameters for these algorithms. An additional limitation of Adam and AdaGrad is that they lack reliable diagnostics. For example, with Adam, the ELBO increases markedly for 'election88' when shifting from $n=16$ to $n=256$, which may suggest that a sample size of $n=256$ is sufficient. However, the gap to the performance of SAA for VI reveals this is not the case; the convergence diagnostics of SAA for VI based on gaps between training and fresh samples are a significant advantage in practice.
>
> Below we provide our experimental results. A condensed version of these results is included in the appendix for quick reference. The comprehensive results can also be accessed via the link provided on OpenReview.
>
> > Theoretical Analysis:
>
> Theoretical convergence results for black-box VI are an important research direction but outside the scope of our paper. The line of work you point to is interesting, but more aligned with traditional black-box VI, which uses first-order stochastic gradient methods. In fact, it has been a complex and long standing question to prove basic convergence results in that setting. One of the first such results [1] was submitted to arxiv concurrently to our own paper. The point of our paper is that an alternative stochastic optimization approach can perform robustly in practice for black-box VI for problems with up to hundreds of latent variables.
>
> [1] Kim, Kyurae, et al. "Black-Box Variational Inference Converges." arXiv preprint arXiv:2305.15349 (2023).
>
> > Limited Scope of Applications:
>
> First, please see our general response https://openreview.net/forum?id=Lvg10LZ5nL&noteId=nJqrGbrF5K
> These models really are representative of challenging inference tasks that arise from real applied statistical modeling. The UCI data sets are representative of real Bayesian logistic regression problems; further, they represent only 6 of the 17 models used in our evaluation, with the others models being a disparate set of real Stan models. It can be seen within our own experimental results that current state of the art methods often perform poorly on these problems, and the general problem of probabilistic inference on such problems is far from being conclusively "solved". Our methods contribute potential solutions to the existing challenges within this space.
>
> Second, the most pertinent claim in our paper is as follows, and we believe we have supported it with clear evidence:
> > “We propose an alternative stochastic optimization approach based on the sample average approximation (SAA) that can be easily made robust for problems involving hundreds of latent variables”

---

> > ### Author Response · Authors · 2023-06-07
> > **Response to review (2/n)**
> >
> > # Comparison of Adam and SAA for VI, n=256
> >
> > | Model         | Adam (ELBO) | SAA for VI (ELBO) | Difference (ELBO) | Adam (Running Time) | SAA for VI (Running Time) | Ratio (Running Time) |
> > |---------------|-------------|-------------------|-------------------|---------------------|----------------------------|----------------------|
> > | **Bayesian log. regr.** |
> > | a1a           | -636.57     | -636.40           | -0.17             | 16.16               | 20.31                      | 0.80                 |
> > | australian    | -256.75     | -256.73           | -0.01             | 12.94               | 4.81                       | 2.69                 |
> > | ionosphere    | -124.36     | -124.35           | -0.00             | 12.41               | 4.33                       | 2.87                 |
> > | madelon       | -2,433.07   | -2,399.65         | -33.42            | 17.03               | 60.31                      | 0.28                 |
> > | mushrooms     | -180.02     | -179.89           | -0.13             | 31.49               | 18.84                      | 1.67                 |
> > | sonar         | -110.09     | -110.04           | -0.05             | 9.81                | 12.48                      | 0.79                 |
> > | **Stan models** |
> > | congress      | 423.59      | 423.55            | 0.05              | 39.40               | 0.82                       | 48.11                |
> > | election88    | -1,446.37   | -1,398.03         | -48.34            | 3,485.68            | 200.34                     | 17.40                |
> > | election88Exp | ---         | -1,381.79         | ---               | ---                 | 83.68                      | ---                  |
> > | electric      | -792.28     | -786.91           | -5.38             | 275.95              | 49.16                      | 5.61                 |
> > | electric-one-pred | -818.00 | -818.01           | 0.01              | 65.63               | 0.62                       | 106.32               |
> > | hepatitis     | -566.51     | -557.36           | -9.14             | 250.90              | 98.69                      | 2.54                 |
> > | hiv-chr       | -77,190.31  | -582.78           | -76,607.53        | 157.79              | 29.91                      | 5.28                 |
> > | irt           | -15,894.76  | -15,884.67        | -10.09            | 62.90               | 109.17                     | 0.58                 |
> > | mesquite      | -29.78      | -29.83            | 0.05              | 39.76               | 0.27                       | 147.35               |
> > | radon         | -1,210.36   | -1,209.46         | -0.90             | 197.40              | 21.19                      | 9.31                 |
> > | wells         | -2,041.90   | -2,041.95         | 0.05              | 17.48               | 0.08                       | 211.03               |

---

> > > ### Author Response · Authors · 2023-06-07
> > > **Response to review (3/n)**
> > >
> > > # Comparison of Adam and SAA for VI, n=1
> > >
> > > | Model         | Adam (ELBO) | SAA for VI (ELBO) | Difference (ELBO) | Adam (Running Time) | SAA for VI (Running Time) | Ratio (Running Time) |
> > > |---------------|-------------|-------------------|-------------------|---------------------|---------------------------|----------------------|
> > > | **Bayesian log. regr.** |
> > > | a1a           | -640.97     | -636.40           | -4.57             | 57.90               | 19.05                     | 3.04                 |
> > > | australian    | -257.15     | -256.73           | -0.42             | 29.89               | 4.81                      | 6.22                 |
> > > | ionosphere    | -124.77     | -124.35           | -0.41             | 20.67               | 4.33                      | 4.78                 |
> > > | madelon       | -4,030.36   | -2,399.65         | -1,630.71         | 633.49              | 58.52                     | 10.82                |
> > > | mushrooms     | -183.96     | -179.89           | -4.07             | 66.70               | 16.27                     | 4.10                 |
> > > | sonar         | -111.69     | -110.04           | -1.65             | 37.19               | 11.82                     | 3.15                 |
> > > | **Stan models** |
> > > | congress      | 423.48      | 423.55            | -0.06             | 74.12               | 0.82                      | 90.50                |
> > > | election88    | ---         | -1,398.03         | ---               | ---                 | 94.75                     | ---                  |
> > > | election88Exp | ---         | -1,381.79         | ---               | ---                 | 38.07                     | ---                  |
> > > | electric      | -1,075.57   | -786.91           | -288.66           | 256.22              | 42.14                     | 6.08                 |
> > > | electric-one-pred | -818.01 | -818.01           | -0.00             | 70.77               | 0.62                      | 114.64               |
> > > | hepatitis     | ---         | -557.36           | ---               | ---                 | 92.46                     | ---                  |
> > > | hiv-chr       | ---         | -582.78           | ---               | ---                 | 29.74                     | ---                  |
> > > | irt           | -16,560.79  | -15,884.67        | -676.13           | 504.18              | 89.94                     | 5.61                 |
> > > | mesquite      | -29.79      | -29.83            | 0.04              | 66.81               | 0.27                      | 247.63               |
> > > | radon         | -4,715.77   | -1,209.46         | -3,506.31         | 141.44              | 14.70                     | 9.62                 |
> > > | wells         | -2,041.90   | -2,041.95         | 0.05              | 24.69               | 0.08                       | 298.10               |

---

> > > > ### Author Response · Authors · 2023-06-07
> > > > **Response to review (4/4)**
> > > >
> > > > # Comparison of AdaGrad and SAA for VI, n=16
> > > >
> > > > | Model             | AdaGrad (ELBO) | SAA for VI (ELBO) | Difference (ELBO) | AdaGrad (Running Time) | SAA for VI (Running Time) | Ratio (Running Time) |
> > > > |-------------------|----------------|-------------------|-------------------|------------------------|---------------------------|----------------------|
> > > > | **Bayesian log. regr.** |
> > > > | a1a               | -636.76        | -636.40           | -0.36             | 12.67                  | 20.31                     | 0.62                 |
> > > > | australian        | -256.77        | -256.73           | -0.04             | 4.74                   | 4.81                      | 0.99                 |
> > > > | ionosphere        | -124.39        | -124.35           | -0.04             | 2.95                   | 4.33                      | 0.68                 |
> > > > | madelon           | -2,469.59      | -2,399.65         | -69.94            | 85.49                  | 59.45                     | 1.44                 |
> > > > | mushrooms         | -181.48        | -179.89           | -1.59             | 68.22                  | 17.30                     | 3.94                 |
> > > > | sonar             | -110.19        | -110.04           | -0.14             | 3.73                   | 12.17                     | 0.31                 |
> > > > | **Stan models** |
> > > > | congress          | 413.88         | 423.55            | -9.66             | 18.36                  | 0.82                      | 22.41                |
> > > > | election88        | ---            | -1,398.03         | ---               | ---                    | 97.20                     | ---                  |
> > > > | election88Exp     | ---            | -1,381.79         | ---               | ---                    | 79.99                     | ---                  |
> > > > | electric          | ---            | -786.91           | ---               | ---                    | 28.36                     | ---                  |
> > > > | electric-one-pred | -5,572.18      | -818.01           | -4,754.17         | 32.53                  | 0.62                      | 52.70                |
> > > > | hepatitis         | ---            | -557.36           | ---               | ---                    | 92.46                     | ---                  |
> > > > | hiv-chr           | ---            | -582.78           | ---               | ---                    | 29.74                     | ---                  |
> > > > | irt               | -15,900.00     | -15,884.67        | -15.33            | 106.84                 | 109.17                    | 0.98                 |
> > > > | mesquite          | -75.93         | -29.83            | -46.10            | 39.74                  | 0.27                      | 147.28               |
> > > > | radon             | ---            | -1,209.46         | ---               | ---                    | 14.70                     | ---                  |
> > > > | wells             | -2,041.90      | -2,041.95         | 0.05              | 7.18                   | 0.08                      | 86.66                |

---

> > ### Comment · Reviewer_D3CN · 2023-06-19
> > **Thank you for your responses to my review.**
> >
> > Thank you for your responses to my review and for addressing the points I raised. I appreciate the effort you have made to conduct additional experiments comparing your method with different settings for Adam and AdaGrad. In line with the feedback from the other reviewers, a more thorough experimental comparison, specifically including other second-order methods and their efficient application in large-scale practical settings, would strengthen the paper further.

---

### Author Response · Authors · 2023-06-07
**Preamble to rebuttal**

It may be helpful to provide some context regarding our problem setting (we have also revised the introduction accordingly): The goal of VI is probabilistic inference: To approximate the distribution of latent variables given observed data and a user-specified model. This is of immense practical interest in fields like astrophysics, epidemiology, political science, psychology, ecology, etc. Typically this is used in applications where the data are limited so that user-provided assumptions are critical to obtain insight from data. However, it is an extremely challenging problem in general (#P-hard, “worse” than NP-hard). All this means that problems with modest dimensionality (e.g. 500) are both (1) of large practical significance and (2) are unsolved with current methods—there is no known software or algorithm that can reliably solve all probabilistic inference methods with less than 500 dimensions. Bayesian inference is a classic problem in machine learning. There are many active projects, often with core contributors from large companies involved in AI (e.g. https://www.tensorflow.org/probability, https://num.pyro.ai/, https://www.pymc.io/, https://mc-stan.org/) Most of the models we use are in posteriorDB (https://github.com/stan-dev/posteriordb), a corpus of problems designed for benchmarking probabilistic inference methods.

Our responses are guided by the TMLR acceptance criteria: (Q1) are the claims supported by evidence? (Q2) would some of TMLR’s audience be interested in the findings? We have made light revisions to motivate the paper to a broader audience and responded to requested changes that are within the paper’s scope, such as discussing hyperparameters or comparing to additional variants of baseline algorithms. However, as our target audience is researchers in approximate inference as it relates to statistical modeling, we have not substantially changed the focus from what is appropriate for this audience.

---

### Decision · Action_Editors · 2023-07-01

**Recommendation:** Reject

**Comment:**

This paper proposes a second-order method for black-box variational inference. The key idea is to improve the accuracy and efficiency of the optimization by solving a sequence of sample average approximations. The proposed approach is evaluated on many datasets and shown to be competitive with state-of-the-art optimizers like Adam.

None of the reviewers support acceptance of this paper. The main reasons are experiments:

* All experiments are small scale and not relevant for large-scale optimization in machine learning today.

* More baselines are needed, such as different variants of Adam, AdaGrad, other second-order methods, and other AutoML methods.

This paper is not aligned with the current trend of large-scale optimization. That being said, this is also not an acceptance criterion for TMLR. However, the paper should be supported by evidence and have audience. Since all experiments in the paper are small scale, I believe that the audience would be small. Therefore, the paper is rejected for now, and we encourage the authors to resubmit. I would love to oversee a revision of the paper with large-scale experiments.

**Audience:**

Not many. Since all experiments in the paper are small scale, I believe that the audience would be small.

**Claims And Evidence:**

Yes. The paper does a decent job in showing improvements on small-scale problems.

**Resubmission Of Major Revision:**

The authors may consider submitting a major revision at a later time.